# On Densest $k$-Subgraph Mining and Diagonal Loading: Optimization Landscape and Finite-Step Exact Convergence Analysis

Qiheng Lu [1]   Nicholas D. Sidiropoulos [1]   Aritra Konar [2]

## Abstract

The Densest $k$-Subgraph (D$k$S) is a fundamental combinatorial problem known for its theoretical hardness and breadth of applications. Recently, Lu et al. (AAAI 2025) introduced a penalty-based non-convex relaxation that achieves promising empirical performance; however, a rigorous theoretical understanding of its success remains unclear. In this work, we bridge this gap by providing a comprehensive theoretical analysis. We first establish the tightness of the relaxation, ensuring that the global maximum values of the original combinatorial problem and the relaxed problem coincide. Then we reveal the benign geometry of the optimization landscape by proving a strict dichotomy of stationary points: all integral stationary points are local maximizers, whereas all non-integral stationary points are strict saddles with explicit positive curvature. We propose a saddle-escaping Frank–Wolfe algorithm and prove that it achieves exact convergence to an integral local maximizer in a finite number of steps.

## 1. Introduction

Extracting densely connected vertex subsets in large graphs is a fundamental task in graph mining (see Lanciano et al. (2024) and references therein), with applications ranging from social media analysis (Dourisboure et al., 2007; Angel et al., 2014) and computational biology (Fratkin et al., 2006; Saha et al., 2010), to financial fraud detection (Hooi et al., 2016; Chen & Tsourakakis, 2022).

A classic and widely employed formulation is the Densest Subgraph (DSG) problem (Goldberg, 1984), which finds the subgraph that maximizes the average induced degree. DSG admits an exact polynomial-time solution via maximum flow (Goldberg, 1984), and an efficient 2-approximation via greedy peeling (Charikar, 2000). Recently, significant progress has been achieved via multi-stage extensions of greedy peeling (Boob et al., 2020; Chekuri et al., 2022) or advanced convex optimization techniques (Danisch et al., 2017; Harb et al., 2022; 2023; Nguyen & Ene, 2024).

However, it has been observed that DSG algorithms tend to extract large and sparsely connected subgraphs in practice (Tsourakakis et al., 2013). This is because the average degree metric is inherently biased towards larger subgraphs: for any subgraph of size $k$, its average degree is bounded above by $k-1$. This bias makes DSG algorithms often overlook smaller, highly cohesive structures (e.g., small cliques) in favor of larger, sparser subgraphs.

These limitations can be addressed by considering the Densest $k$-Subgraph (D$k$S) problem (Feige et al., 2001), which explicitly seeks a subgraph of size exactly $k$ with maximum density. Other variants of cardinality-constrained dense subgraph problems have also been explored, such as the Densest at-least-$k$ Subgraph (Dal$k$S) problem and the Densest at-most-$k$ Subgraph (Dam$k$S) problem (Andersen & Chellapilla, 2009; Khuller & Saha, 2009). However, in contrast to D$k$S, such variants cannot explore dense subgraphs across the full range of discrete sizes $k$. Moreover, Dal$k$S often exhibits the same bias as DSG, potentially yielding subgraphs that significantly exceed the threshold $k$.

Imposing the cardinality constraint renders D$k$S NP-hard (Feige et al., 2001). Moreover, D$k$S is known to be difficult to approximate (Khot, 2006; Manurangsi, 2017; Jones et al., 2023) and the best known polynomial-time algorithm achieves only an $O(n^{1/4+\epsilon})$ approximation ratio in time $n^{O(1/\epsilon)}$ for every $\epsilon > 0$ (Bhaskara et al., 2010).

To circumvent these barriers, significant efforts have turned to relaxation techniques. However, theoretical frameworks ranging from hierarchical semidefinite programming (SDP) relaxations (Bhaskara et al., 2012) to quasi-polynomial schemes (Barman, 2018) to convex relaxations based on matrix lifting (Ames, 2015; Bombina & Ames, 2020) typically incur prohibitive computational costs. This stems from the quadratic expansion in the number of variables in

---

[1] University of Virginia [2] KU Leuven. Correspondence to: Qiheng Lu <luqh@virginia.edu>, Nicholas D. Sidiropoulos <nikos@virginia.edu>, Aritra Konar <aritra.konar@kuleuven.be>.

*Proceedings of the $43^{rd}$ International Conference on Machine Learning*, Seoul, South Korea. PMLR 306, 2026. Copyright 2026 by the author(s).

matrix lifting and the cubic complexity of Singular Value Decomposition (SVD).

To approximate D$k$S, Papailiopoulos et al. (2014) proposed the Spannogram framework, which exhibits polynomial-time solvability for constant-rank adjacency matrices. However, its complexity scales exponentially with the rank (e.g., $O(n^3)$ for rank-2), limiting its scalability on large graphs. Alternatively, Yuan & Zhang (2013) leverage the Truncated Power Method (TPM) to tackle D$k$S. While TPM provides convergence guarantees under specific conditions, its effectiveness is often compromised by a tendency to converge to sub-optimal results (Papailiopoulos et al., 2014).

More recently, gradient-based algorithms for tackling continuous relaxations of D$k$S have been proposed. Hager et al. (2016) introduced a pair of algorithms for solving a relaxation based on sparse Principal Component Analysis (PCA), while Sotirov (2020) applied coordinate descent on a continuous relaxation of D$k$S, obtained by relaxing the feasible set to its convex hull. Konar & Sidiropoulos (2021) convexify the objective function through the Lovász extension and relax the feasible set to its convex hull, leveraging the Alternating Direction Method of Multipliers (ADMM) to solve the resulting formulation. However, none of the aforementioned works (Hager et al., 2016; Sotirov, 2020; Konar & Sidiropoulos, 2021) provide a theoretical characterization of the integrality gap. This lack of analysis makes it difficult to assess the tightness of the proposed relaxations or to determine the extent of the gap when the relaxations are not exact.

Recently, Liu et al. (2024; 2026) proposed two penalty-based *exact* non-convex, continuous relaxations of D$k$S and leveraged gradient-based algorithms embedded within a homotopy-based framework to solve them. However, the exactness guarantees of these formulations rely on conservative sufficient conditions on the penalty parameter $\lambda$. Specifically, Liu et al. (2026) requires $\lambda = \Omega(\sqrt{n}\|\boldsymbol{A}\|_2)$ ($n$ is the number of vertices and $\boldsymbol{A}$ is the adjacency matrix) to ensure tightness, while Liu et al. (2024) requires $\lambda = \Omega(\|\boldsymbol{A}\|_2)$. Without investigating the problem's intrinsic geometric properties, such parameter choices may result in challenging optimization landscapes for gradient-based algorithms. Consequently, their empirical performance can be sensitive to homotopy-related hyperparameters, occasionally requiring dataset-specific configurations to consistently obtain high-quality solutions. In addition, none of (Liu et al., 2024; 2026) characterize the stationary points of their relaxed problems, which hinders the development of more principled algorithmic designs.

Closest to our present work, Lu et al. (2025) proposed a non-convex continuous relaxation of D$k$S based on a quadratic penalty formulation, and investigated the role of the penalty parameter $\lambda$ in guaranteeing an exact relaxation

of D$k$S. Conventional wisdom suggests that a large, *graph-dependent* penalty parameter $\lambda \geq -\lambda_{\min}(\boldsymbol{A})$ is needed for exactness, where $\lambda_{\min}(\boldsymbol{A})$ is the smallest eigenvalue of the adjacency matrix. By leveraging an extension of the classic Motzkin-Straus theorem (Motzkin & Straus, 1965), they established the surprising result that the small *graph-independent* value of $\lambda = 1$ is both necessary and sufficient to guarantee a tight relaxation when the subgraph size $k$ is smaller than the maximum clique size $\omega(\mathcal{G})$. They also empirically validated that $\lambda = 1$ results in a benign non-convex optimization landscape, thereby enabling gradient-based methods to find high-quality solutions for D$k$S. In particular, Lu et al. (2025) demonstrated that selecting $\lambda = 1$ and solving the relaxation via the Frank–Wolfe algorithm (Frank & Wolfe, 1956; Jaggi, 2013; Braun et al., 2025) effectively balances the quality of the solution and the convergence speed compared to both larger and smaller values of $\lambda$. Furthermore, their algorithm also exhibited state-of-the-art performance in terms of obtaining high-quality solutions for D$k$S while being scalable to large datasets.

Despite these successes, the framework proposed by Lu et al. (2025) leaves several important theoretical questions unresolved:

• **Tightness Analysis:** The current guarantee for a tight relaxation is limited to subgraph sizes $k \leq \omega(\mathcal{G})$, which is generally a small interval for real graphs. For larger subgraph sizes $k > \omega(\mathcal{G})$, it remains unclear whether their relaxation preserves tightness when $\lambda = 1$.

• **Landscape Analysis:** While $\lambda = 1$ is empirically observed to be effective in finding high-quality solutions for D$k$S, a rigorous characterization of the optimization landscape, which can help explain these successes, is lacking.

• **Convergence Analysis of Frank–Wolfe:** We also observe that their algorithm always converges to an integral stationary point, thereby eliminating the need for any post-processing rounding procedure. Currently, there is no theoretical explanation for such behavior.

**Contributions:** We resolve these open questions regarding the framework of Lu et al. (2025), improving the understanding of its effectiveness for D$k$S. Our main contributions are summarized as follows:

• **Tightness Analysis:** We prove that the relaxation is tight for *arbitrary* subgraph sizes $k$ provided that the penalty parameter $\lambda \geq 1$ (Theorem 2.3). This result affirmatively answers the open question regarding tightness in the regime $k > \omega(\mathcal{G})$, extending the validity of the relaxation to general subgraph mining scenarios.

• **Landscape Analysis:** We present the first rigorous geometric characterization of the relaxation in Lu et al. (2025), clarifying its effectiveness for D$k$S. Specifically, we prove

a strict dichotomy of the set of stationary points: all integral stationary points are local maximizers (Theorem 3.7), whereas all non-integral stationary points are strict saddle points when $\lambda > 1$ is non-integral (Theorem 3.10). Furthermore, by establishing the monotonicity of the set of local maximizers with respect to $\lambda$ (Theorem 3.2), we offer a theoretical insight for the empirical observation that relatively smaller $\lambda$ yields better solution quality.

• **Convergence Analysis:** We resolve the theoretical mystery of why Frank–Wolfe yields integral solutions without rounding. We propose a saddle-escaping variant of Frank–Wolfe that leverages explicit strict ascent directions to escape saddle points, in contrast to existing techniques such as random perturbation (surveyed in Section 6). We prove that our algorithm escapes non-integral saddle points and achieves *finite-step exact convergence* to an integral local maximizer, thereby explaining the algorithm's natural convergence to integral solutions without rounding.

## 2. Problem Formulation and Tightness Analysis

Consider an unweighted, undirected, and simple (no self-loops or multiple edges) graph $\mathcal{G} := (\mathcal{V}, \mathcal{E})$ with $n$ vertices and $m$ edges, where $\mathcal{V} := \{1, 2, \ldots, n\}$ is the vertex set and $\mathcal{E}$ is the edge set. D$k$S can be formulated as the following binary quadratic maximization problem

$$\max_{\boldsymbol{x} \in \{0,1\}^n} \quad f(\boldsymbol{x}) = \frac{1}{2}\boldsymbol{x}^\top \boldsymbol{A}\boldsymbol{x}$$
$$\text{s.t.} \quad \sum_{i \in [n]} x_i = k, \tag{1}$$

where $\boldsymbol{A}$ is the symmetric $n \times n$ adjacency matrix of $\mathcal{G}$ and $\boldsymbol{x}$ is a binary indicator vector.

By leveraging diagonal loading, Lu et al. (2025) equivalently recast (1) into

$$\max_{\boldsymbol{x} \in \{0,1\}^n} \quad g(\boldsymbol{x}) = \frac{1}{2}\boldsymbol{x}^\top (\boldsymbol{A} + \lambda \boldsymbol{I})\boldsymbol{x}$$
$$\text{s.t.} \quad \sum_{i \in [n]} x_i = k, \tag{2}$$

where $\lambda > 0$ is a penalty parameter. Relaxing the feasible set of (2) to its convex hull yields the following continuous formulation:

$$\max_{\boldsymbol{x} \in [0,1]^n} \quad g(\boldsymbol{x}) = \frac{1}{2}\boldsymbol{x}^\top (\boldsymbol{A} + \lambda \boldsymbol{I})\boldsymbol{x}$$
$$\text{s.t.} \quad \sum_{i \in [n]} x_i = k. \tag{3}$$

Throughout this paper, we denote the feasible set of (3)

by $\mathcal{C}_k^n$. Similar to Lu et al. (2025), we say that the relaxation from (2) to (3) is *tight* if their global maximum values coincide, or equivalently, if any global maximizer of (2) remains optimal for (3). Regarding tightness, we first review the main theoretical result in Lu et al. (2025).

**Theorem 2.1** (Corollary of Theorem 2 in Lu et al. (2025)). *If $\lambda < 1$, the relaxation from (2) to (3) is not tight for a given subgraph size $k < \omega(\mathcal{G})$, where $\omega(\mathcal{G})$ is the maximum clique size of $\mathcal{G}$.*

Theorem 2.1 shows that selecting the penalty parameter $\lambda \geq 1$ is a necessary condition for the general tightness of the relaxation from (2) to (3). We now prove that this condition is also sufficient for *every* subgraph size $k$.

**Theorem 2.2** (Lemma 5.1 in Barman (2018)). *If $\lambda = 2$, the relaxation from (2) to (3) is tight for any $k$.*

**Theorem 2.3** (Extension of Lemma 5.1 in Barman (2018)). *If $\lambda \geq 1$, the relaxation from (2) to (3) is tight for any $k$.*

*Proof.* Please refer to Appendix A. □

*Remark* 2.4. Theorem 2.3 guarantees that for $\lambda \geq 1$, the optimal objective values of the relaxation (3) and the original problem (2) coincide. However, this does not imply that the sets of optimal solutions are identical. Specifically, when $\lambda = 1$, the relaxed problem (3) may admit non-integral global maximizers in addition to the binary optimal solutions of (2). Choosing $\lambda > 1$ is sufficient to ensure that the sets of optimal solutions are the same because $\|\boldsymbol{x}\|_2^2$ is maximized over the feasible set of (3) if and only if $\boldsymbol{x}$ is integral.

Theorem 2.3 establishes the tightness of the relaxation (3) for any $k$ when $\lambda \geq 1$. However, since the optimization problem (3) is non-convex in general, gradient-based algorithms may get trapped at stationary points, depending on the optimization landscape. Therefore, guaranteeing tightness alone is insufficient to explain the empirical successes observed by Lu et al. (2025). In the next section, we analyze the influence of $\lambda$ on the optimization landscape of (3).

## 3. Analyzing the Optimization Landscape

In this section, we analyze the geometry of the relaxation (3) to characterize its evolution as a function of $\lambda$ and reveal a benign optimization landscape, despite non-convexity, for small $\lambda > 1$.

### 3.1. Monotonicity of Local Maximizers

We begin with the local maximizers of (3), for which we prove the following pair of results.

**Lemma 3.1.** *If $\lambda > 1$, then all local maximizers of (3) are integral.*

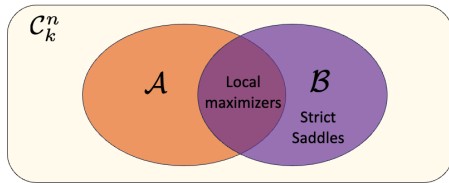

*Figure 1.* Visual representation of the main results in Section 3. Yellow rectangle: set of all feasible points $\mathcal{C}_k^n$ of (3). Orange set $\mathcal{A}$: subset of all integral points of (3). Purple set $\mathcal{B}$: subset of all stationary points of (3). **Theorem 3.7:** $\mathcal{A} \cap \mathcal{B}$ is the set of all local maximizers of (3) when $\lambda > 1$ is non-integral. **Theorem 3.10:** $\mathcal{B} \setminus \mathcal{A}$ is the set of all strict saddle points of (3) when $\lambda > 1$ is non-integral.

*Proof.* Please refer to Appendix B.  □

**Theorem 3.2** (Monotonicity of Local Maximizers). *If $\lambda_2 > \lambda_1 > 1$ and $\boldsymbol{x}$ is a local maximizer of (3) with the penalty parameter $\lambda_1$, then $\boldsymbol{x}$ is also a local maximizer of (3) with the penalty parameter $\lambda_2$.*

*Proof.* Please refer to Appendix C.  □

Theorem 3.2 demonstrates that for a given $\lambda > 1$, the set of local maximizers of (3) is contained within that of a larger $\lambda$. We conclude that increasing $\lambda$ can only proliferate, but never remove local maximizers, thereby resulting in a more complicated optimization landscape. This suggests that choosing a smaller value of $\lambda$ in (3) results in a more "friendly" optimization landscape for gradient-based algorithms, provided that $\lambda > 1$. Additionally, it also provides a theoretical insight for the empirical observation in Lu et al. (2025) that increasing $\lambda$ often results in Frank–Wolfe getting trapped in low-quality integral solutions corresponding to sparsely connected subgraphs.

### 3.2. Characterization of Stationary Points

In this subsection, we rigorously characterize the stationary points of (3), which is crucial for unveiling the underlying benign nature of the optimization landscape.

To facilitate our analysis, we first introduce necessary notation. Let $\boldsymbol{v}(\boldsymbol{x}) := \nabla g(\boldsymbol{x}) = (\boldsymbol{A} + \lambda \boldsymbol{I})\boldsymbol{x}$ denote the gradient vector. For any $\boldsymbol{x}$ that is a feasible point of (3), we partition the index set $[n] := \{1, 2, \ldots, n\}$ into three disjoint sets: $\mathcal{S}_0(\boldsymbol{x}) := \{i \in [n] \mid x_i = 0\}$, $\mathcal{S}_f(\boldsymbol{x}) := \{i \in [n] \mid 0 < x_i < 1\}$, and $\mathcal{S}_1(\boldsymbol{x}) := \{i \in [n] \mid x_i = 1\}$. For brevity, we will omit the argument $\boldsymbol{x}$ (e.g., writing $\mathcal{S}_f$ instead of $\mathcal{S}_f(\boldsymbol{x})$) when the context is clear. Additionally, we let $L := \|\boldsymbol{A} + \lambda \boldsymbol{I}\|_2$ denote the spectral norm of the Hessian, which serves as the Lipschitz constant of the gradient $\boldsymbol{v}(\boldsymbol{x})$.

**Definition 3.3** (Stationary Point). *$\boldsymbol{x}$ is a stationary point of (3) if the following first-order condition holds: $\boldsymbol{v}(\boldsymbol{x})^\top (\boldsymbol{y} - \boldsymbol{x}) \leq 0$ for every $\boldsymbol{y}$ feasible for (3).*

Since the constraints of (3) satisfy the Linearity Constraint Qualification (LCQ), the Karush–Kuhn–Tucker (KKT) conditions also equivalently characterize the stationary points of (3). Using these conditions, we provide an explicit characterization of the stationary points.

**Theorem 3.4** (Necessary and Sufficient Conditions for Stationarity). *$\boldsymbol{x}$ is a stationary point of (3) if and only if there exists a scalar $\mu$ such that $v_i \leq \mu$ for every $i \in \mathcal{S}_0$, $v_i \geq \mu$ for every $i \in \mathcal{S}_1$, and $v_i = \mu$ for every $i \in \mathcal{S}_f$.*

*Proof.* Please refer to Appendix D.  □

As a corollary of Theorem 3.4, we obtain the following characterization of the integral stationary points of (3).

**Corollary 3.5.** *An integral point $\boldsymbol{x}$ is a stationary point of (3) if and only if $\max_{i \in \mathcal{S}_0} v_i \leq \min_{i \in \mathcal{S}_1} v_i$.*

Intuitively, the condition in Corollary 3.5 implies that for an integral stationary point, the marginal gain (gradient) of any vertex selected in the size-$k$ subgraph must be no less than that of any unselected vertex.

Next, inspired by the work of Hager & Krylyuk (1999) on graph partitioning, we establish a link between the sets of local maximizers (Lemma 3.1) and the integral stationary points of (3) (Corollary 3.5). To this end, the following stepping-stone result is key, as it establishes the necessary and sufficient conditions for characterizing local optimality.

**Theorem 3.6** (Necessary and Sufficient Conditions for Local Optimality). *When $\lambda > 1$, $\boldsymbol{x}$ is a local maximizer of (3) if and only if $\boldsymbol{x}$ is integral and $\max_{i \in \mathcal{S}_0} v_i < \min_{i \in \mathcal{S}_1} v_i$.*

*Proof.* Please refer to Appendix E.  □

Using the above conditions, we arrive at our first major result of this subsection.

**Theorem 3.7.** *All integral stationary points of (3) are local maximizers when $\lambda > 1$ is non-integral.*

*Proof.* By Corollary 3.5 and Theorem 3.6, we know that an integral stationary point is not a local maximizer if and only if $\max_{i \in \mathcal{S}_0} v_i = \min_{i \in \mathcal{S}_1} v_i$. Since $\lambda$ is non-integral, $\min_{i \in \mathcal{S}_1} v_i$ is also non-integral and therefore cannot equal the integer $\max_{i \in \mathcal{S}_0} v_i$.  □

*Remark* 3.8. Lemma 3.1 and Theorem 3.7 establish a significant algorithmic implication: provided we have an algorithm that converges to a stationary point, certifying local optimality takes only $O(n)$ time via an integrality check.

Having fully characterized the integral stationary points, we now proceed to analyze the non-integral ones. We first introduce the definition of a strict saddle point.

**Definition 3.9** (Strict Saddle Point). A stationary point $x$ is a strict saddle point of (3) if the following first and second-order conditions hold: there exists a feasible $y$ for (3) such that $\nabla g(x)^\top(y-x) = 0$ and $(y-x)^\top \nabla^2 g(x)(y-x) > 0$.

With this definition in hand, we arrive at our second major result of this subsection.

**Theorem 3.10.** *All non-integral stationary points of* (3) *are strict saddle points when* $\lambda > 1$.

*Proof.* Please refer to Appendix F.  □

Collectively, Theorems 3.7 and 3.10 establish a strict dichotomy in the optimization landscape: when $\lambda > 1$ is non-integral, every stationary point is either an integral local maximizer or a non-integral strict saddle point with explicit positive curvature. This confirms the benign landscape of (3), serving as a vital theoretical foundation for the design of effective algorithms.

*Remark* 3.11. Although Theorem 3.10 requires $\lambda > 1$, it nonetheless provides insight into the good empirical results obtained with $\lambda = 1$ in Lu et al. (2025). In particular, Theorem 3.4 establishes necessary and sufficient conditions for stationary points, which holds for $\lambda = 1$ as well. Consider a non-integral stationary point under $\lambda = 1$. For such a point to have no strict ascent direction, we need $a_{ij} = 1$ for every pair of distinct indices $i, j \in \mathcal{S}_f$, that is, every pair of distinct indices in the non-integral index set must be connected by an edge. Otherwise, the quadratic term remains positive, implying the existence of a strict ascent direction. Given that real-world graphs are typically sparse, such stationary points are typically strict saddle points even when $\lambda = 1$, making it unlikely for the first-order algorithms to be trapped in their vicinity.

# 4. Saddle-Escaping Frank–Wolfe: Algorithm and Convergence Analysis

Motivated by the benign landscape of (3), we propose the Saddle-Escaping Frank–Wolfe (SE-FW) algorithm in this section. The core idea is to augment the classic Frank–Wolfe method (Frank & Wolfe, 1956; Jaggi, 2013) with an escape step that is triggered specifically in the neighborhood of saddle points. The literature related to saddle point escape is reviewed in Section 6.

**Roadmap of Section 4:** We first prove finite-step exact convergence once the iterate enters the basin of an integral local maximizer (Theorem 4.1). We then show that both standard Frank–Wolfe steps (Theorem 4.12) and escape steps (Corollary 4.9) strictly increase the objective. Based on these results, we bound the total number of escape steps (Corollary 4.13). Finally, leveraging the convergence guarantee of the standard Frank–Wolfe steps (Theorem 4.12),

---

**Algorithm 1** Saddle-Escaping Frank–Wolfe for (3)

**Require:** The adjacency matrix $A$, the subgraph size $k$, and the non-integral penalty parameter $\lambda > 1$. Let $x^{(0)}$ be a feasible initialization.

1: $L \leftarrow \|A + \lambda I\|_2$
2: $t \leftarrow 0$
3: **while** the Frank–Wolfe gap $G(x^{(t)}) = \max_{s \in \mathcal{C}_k^n} \langle \nabla g(x^{(t)}), s - x^{(t)} \rangle$ is not zero **do**
4: $\quad v^{(t)} \leftarrow (A + \lambda I)x^{(t)}$
5: $\quad s^{(t)} \leftarrow 0$
6: $\quad s^{(t)}[\text{top}_k(v^{(t)})] \leftarrow 1$
7: $\quad d^{(t)} \leftarrow s^{(t)} - x^{(t)}$
8: $\quad \gamma^{(t)} \leftarrow \min\left\{1, \frac{(v^{(t)})^\top d^{(t)}}{L\|d^{(t)}\|_2^2}\right\}$
9: $\quad$ **if** the Frank–Wolfe gap is smaller than the threshold discussed in Remark 4.11, and either $s^{(t)}$ is not a local maximizer, or $s^{(t)}$ is a local maximizer but $x^{(t)}$ is not within the neighborhood of $s^{(t)}$ derived in Theorem 4.1 **then**
10: $\quad\quad$ Let $j, l$ be indices of the $k$-th and $(k+1)$-th largest elements of $x^{(t)}$.
11: $\quad\quad$ **if** $v_j^{(t)} \geq v_l^{(t)}$ **then**
12: $\quad\quad\quad \delta^{(t)} \leftarrow \min\{x_l, 1 - x_j\}$
13: $\quad\quad\quad x^{(t+1)} \leftarrow x^{(t)} + \delta^{(t)}(e_j - e_l)$
14: $\quad\quad$ **else**
15: $\quad\quad\quad \delta^{(t)} \leftarrow \min\{x_j, 1 - x_l\}$
16: $\quad\quad\quad x^{(t+1)} \leftarrow x^{(t)} + \delta^{(t)}(e_l - e_j)$
17: $\quad\quad$ **end if**
18: $\quad$ **else**
19: $\quad\quad x^{(t+1)} \leftarrow x^{(t)} + \gamma^{(t)} d^{(t)}$
20: $\quad$ **end if**
21: $\quad t \leftarrow t + 1$
22: **end while**

---

we obtain a finite-step exact convergence result.

## 4.1. Algorithm Description

The SE-FW algorithm (Algorithm 1) is designed to identify integral local maximizers with finite-step exact convergence. Unlike many existing non-convex solvers that rely on random perturbations to escape saddle points (Ge et al., 2015; Jin et al., 2017; Lu et al., 2020), SE-FW exploits the explicit positive curvature of the landscape to construct deterministic directions of strict ascent. The algorithm operates via a hybrid strategy, alternating between standard Frank–Wolfe updates (Lines 4–8 and 19) and specific operations to escape strict saddle points (Lines 9–17).

The standard Frank–Wolfe updates consist of computing the gradient of $g(x^t)$ (Line 4), followed by computing the solution of the linear maximization oracle (LMO)

$$s^{(t)} \in \arg\max_{s \in \mathcal{C}_k^n}(v^{(t)})^\top s, \qquad (4)$$

where $\mathcal{C}_k^n$ denotes the constraint set of (3). (4) has a closed-form solution (Lines 5–6), which in turn yields the ascent direction $\boldsymbol{d}^{(t)}$ (Line 7). The adaptive step-size rule is specified in Line 8 (Bertsekas, 2016, p. 268). The saddle escape criterion (Line 9) is designed to detect when the algorithm stagnates in the neighborhood of saddle points. Specifically, the escape step is triggered only when the Frank–Wolfe gap is smaller than a threshold and $\boldsymbol{x}^{(t)}$ is not within the basin of attraction of a local maximizer, which can be certified in $O(m+n)$ time following Theorem 3.6. A more detailed discussion of this criterion can be found in Remark 4.11. The algorithm terminates when the Frank–Wolfe gap is zero, implying exact convergence to a stationary point (specifically, a local maximizer due to the saddle escape mechanism).

A key computational advantage of SE-FW is its low per-iteration cost, dominated by sparse matrix-vector multiplication (SpMV) in Lines 4 and 9, and linear-time selection operations (via Quickselect) in Lines 6 and 10. Thus, the total complexity per iteration is $O(m + n)$, which is highly efficient for large-scale problems.

## 4.2. Finite-Step Local Convergence

We first consider the algorithm's behavior in the neighborhood of local maximizers.

**Theorem 4.1.** *Let $\boldsymbol{x}$ be a local maximizer of (3) with a non-integral $\lambda > 1$, and let $\sigma_{\boldsymbol{x}} = \min_{i \in \mathcal{S}_1(\boldsymbol{x})} v_i(\boldsymbol{x}) - \max_{i \in \mathcal{S}_0(\boldsymbol{x})} v_i(\boldsymbol{x}) > 0$ denote the gradient gap at $\boldsymbol{x}$. If the iterate $\boldsymbol{x}^{(t)}$ satisfies $\|\boldsymbol{x}^{(t)} - \boldsymbol{x}\|_2 \leq \frac{\sigma_{\boldsymbol{x}}}{4L}$, then the algorithm converges exactly to the local maximizer in the next step, i.e., $\boldsymbol{x}^{(t+1)} = \boldsymbol{x}$.*

*Proof.* Please refer to Appendix G. $\square$

*Remark* 4.2. The exact convergence relies on the adaptive step size $\min\left\{1, \frac{(\boldsymbol{v}^{(t)})^\mathsf{T} \boldsymbol{d}^{(t)}}{L\|\boldsymbol{d}^{(t)}\|_2^2}\right\}$ (Bertsekas, 2016, p. 268). As $\|\boldsymbol{d}^{(t)}\|_2 \to 0$, the quadratic denominator allows the step size to reach 1. In contrast, another well-known adaptive step size $\min\left\{1, \frac{(\boldsymbol{v}^{(t)})^\mathsf{T} \boldsymbol{d}^{(t)}}{2kL}\right\}$ (Lacoste-Julien, 2016) vanishes asymptotically due to the constant denominator, preventing the algorithm from exact convergence.

**Corollary 4.3.** *When $1 < \lambda < 2$, the gradient gap admits a uniform lower bound $\sigma_{\boldsymbol{x}} \geq \lambda - 1$. This implies that $\frac{\lambda-1}{4L}$ serves as a uniform lower bound for the radius of the basin of attraction, independent of the specific local maximizer.*

In summary, these results guarantee finite-step convergence once the iterate enters the basin of attraction of a local maximizer. The remaining challenge lies in ensuring that the algorithm can effectively escape strict saddle points to reach these attractive regions, which is the focus of the next subsection.

## 4.3. Escaping Saddle Points: Ascent Analysis

Suppose that the iterate $\boldsymbol{x}^{(t)}$ reaches a strict saddle point. Without additional intervention, the standard Frank–Wolfe algorithm would terminate at this point. Hence, we propose additional "escape steps" which exploit the positive curvature at $\boldsymbol{x}^{(t)}$ to construct directions of strict ascent to enable the algorithm to escape such points.

To this end, quantifying the ascent of (3) per escape step is crucial. Since strict saddle points are non-integral (Theorem 3.10), the following corollary of Theorem 2.3 applies.

**Corollary 4.4.** *Let $\boldsymbol{x}$ be a strict saddle point of (3) with a non-integral $\lambda > 1$. There exist two distinct non-integral entries $x_j$ and $x_l$ such that $v_j \geq v_l$. Let $\delta = \min\{x_l, 1 - x_j\} > 0$ and the ascent direction $\boldsymbol{d} = \boldsymbol{e}_j - \boldsymbol{e}_l$. Then, the ascent*

$$g(\boldsymbol{x} + \delta\boldsymbol{d}) - g(\boldsymbol{x}) \geq (\lambda - 1)\delta^2 > 0. \qquad (5)$$

The result shows that for any strict saddle point, there exists a direction $\boldsymbol{d}$ which guarantees strict ascent. Hence, if we can compute such directions, they can be used for escape. However, in order to demonstrate sufficient ascent, the dependence of the lower bound (38) on the term $\delta$ needs further clarification. Consequently, our analysis focuses on identifying distinct indices $j$ and $l$ that provide a lower bound on $\min\{x_j, 1 - x_j, x_l, 1 - x_l\}$, which in turn yields a lower bound for $\delta$.

**Theorem 4.5.** *Let $\boldsymbol{x}$ be a strict saddle point of (3) with a non-integral $\lambda > 1$, and let $\boldsymbol{y}$ be any integral feasible point of (3) closest to $\boldsymbol{x}$ (in the Euclidean norm). Then there exist two distinct indices $j$ and $l$ such that $\min\{x_j, 1 - x_j, x_l, 1 - x_l\} \geq \frac{\xi_{\boldsymbol{y}}}{2(\lambda+1)n}$, where $\xi_{\boldsymbol{y}} = |\min_{i \in \mathcal{S}_1(\boldsymbol{y})} v_i(\boldsymbol{y}) - \max_{i \in \mathcal{S}_0(\boldsymbol{y})} v_i(\boldsymbol{y})|$.*

*Proof.* Please refer to Appendix H. $\square$

*Remark* 4.6. Through the proof of Theorem 4.5 in Appendix H, by Lemma H.1, we know that the indices $j \in \arg\min_{i \in \mathcal{S}_1(\boldsymbol{y}) \cap \mathcal{S}_f(\boldsymbol{x})} x_i$ and $l \in \arg\max_{i \in \mathcal{S}_0(\boldsymbol{y}) \cap \mathcal{S}_f(\boldsymbol{x})} x_i$ correspond to the $k$-th largest element $x_{(k)}$ and the $(k + 1)$-th largest element $x_{(k+1)}$, respectively. The average time complexity for finding the $k$-th largest element $x_{(k)}$ and the $(k + 1)$-th largest element $x_{(k+1)}$ is $O(n)$ via Quickselect.

Theorem 4.5 provides a lower bound on $\min\{x_j, 1 - x_j, x_l, 1 - x_l\}$. We note that the bound is order-wise tight in the worst case. Please refer to Appendix I for more details.

Substituting the result of Theorem 4.5 in (38), we obtain the desired sufficient ascent result.

**Corollary 4.7.** *Each escape at a saddle point $\boldsymbol{x}$ increases the objective value at least $\frac{(\lambda-1)\xi_{\boldsymbol{y}}^2}{4(\lambda+1)^2 n^2}$, where $\boldsymbol{y}$ is any integral feasible point of (3) closest to $\boldsymbol{x}$ (in the Euclidean norm) and $\xi_{\boldsymbol{y}} = |\min_{i \in \mathcal{S}_1(\boldsymbol{y})} v_i(\boldsymbol{y}) - \max_{i \in \mathcal{S}_0(\boldsymbol{y})} v_i(\boldsymbol{y})|$.*

While Corollary 4.7 guarantees ascent from exact saddle points, practical implementations rarely land precisely on such points. Instead, the algorithm operates within a neighborhood of saddle points. Motivated by such an observation, we now extend our analysis to bound the objective improvement when escaping from these approximate saddle points.

**Lemma 4.8.** *Let $x, y \in \mathbb{R}^n$, and let $x_{(i)}$ and $y_{(i)}$ denote the $i$-th largest elements of $x$ and $y$, respectively. If $\|x - y\|_\infty \leq \epsilon$, then $|x_{(i)} - y_{(i)}| \leq \epsilon$ for every $i \in [n]$.*

*Proof.* Please refer to Appendix J. □

**Corollary 4.9.** *If the iterate $x^{(t)}$ lies in the $\frac{\xi_y}{4(\lambda+1)n}$-neighborhood of a strict saddle point $x$ with respect to the $\ell^\infty$-norm, where $y$ is any integral feasible point of (3) closest to $x$ (in the Euclidean norm) and $\xi_y = |\min_{i \in \mathcal{S}_1(y)} v_i(y) - \max_{i \in \mathcal{S}_0(y)} v_i(y)|$, then selecting the $k$-th and $(k+1)$-th largest elements of $x^{(t)}$ still guarantees an increase in the objective value $\Omega(1/n^2)$ when $\lambda - 1 = \Theta(1)$ and $2 - \lambda = \Theta(1)$ (e.g., $\lambda = 1.5$).*

Corollary 4.9 establishes a theoretical guarantee for objective ascent within a neighborhood of a saddle point. However, the condition relies on the distance of $x^{(t)}$ to the saddle point, which is unknown during the algorithm's execution. To make this result operationally useful, we need a computable surrogate to gauge proximity to the saddle point. To this end, we employ the Frank–Wolfe gap, a standard metric that is readily available at each iteration.

**Theorem 4.10.** *For the optimization problem (3), there exist two scalars $\tau > 0$ and $\theta > 0$ such that for any feasible point $x$, if $G(x) \leq \theta$, then $\|x - y\|_\infty \leq \|x - y\|_2 \leq \tau \sqrt{G(x)}$, where $G(x)$ is the Frank–Wolfe gap at $x$ and $y$ is any stationary point closest to $x$ (in the Euclidean norm).*

*Proof.* Please refer to Appendix K. □

*Remark* 4.11. $\tau$ and $\theta$ are scalars that depend only on the structure of the optimization problem (3). To guarantee that $x^{(t)}$ lies in the neighborhood mentioned in Corollary 4.9, we require that the Frank–Wolfe gap $G(x^{(t)}) \leq \min\left\{\frac{\xi_y^2}{16(\lambda+1)^2\tau^2 n^2}, \theta\right\}$. Since the exact values of these two scalars are generally unknown, explicitly calculating the threshold for the Frank–Wolfe gap is infeasible. To address this, we employ an adaptive strategy. Specifically, if the objective gain from an escape step does not meet the theoretical bound in Corollary 4.9, we reduce the trigger threshold by a factor $\beta \in (0, 1)$ to enforce stricter proximity.

### 4.4. Global Convergence Analysis

Having established the ascent guarantee during the saddle escape step, we now turn to the global convergence analysis of the proposed algorithm.

**Theorem 4.12.** *Let $\{x^{(t)}\}$ be the sequence generated by Algorithm 1 with no saddle point escape step triggered, then the objective values $\{g(x^{(t)})\}$ are strictly monotonically increasing until a stationary point is reached, and the minimum Frank–Wolfe gap $\tilde{G}_T = \min_{0 \leq t \leq T} G(x^{(t)})$ satisfies:*

$$
\tilde{G}_T \leq \begin{cases} \frac{2h_0}{T+1} & \text{if } \tilde{G}_T > 2kL, \\ \sqrt{\frac{4h_0 kL}{T+1}} & \text{if } \tilde{G}_T \leq 2kL. \end{cases} \tag{6}
$$

*where $h_0 = \max_{x \in \mathcal{C}_k^n} g(x) - g(x^{(0)})$ and $\mathcal{C}_k^n = \{y \in [0,1]^n \mid \sum_{i \in [n]} y_i = k\}$. This implies a convergence rate of $O(1/\sqrt{t})$ to a stationary point.*

*Proof.* Please refer to Appendix L. □

By combining the strict monotonicity ensured by Theorem 4.12 for standard Frank–Wolfe steps with the $\Omega(1/n^2)$ objective increase guaranteed by Corollary 4.9 for escape steps, together with the $O(k^2)$ upper bound on the objective value, we obtain an upper bound on the number of escapes.

**Corollary 4.13.** *The number of saddle escapes is at most $O(k^2 n^2)$ when $\lambda - 1 = \Theta(1)$ and $2 - \lambda = \Theta(1)$.*

Synthesizing the finite-step local convergence, the strict monotonicity in objective value, the convergence rate of Frank–Wolfe, and the bound on the number of escape steps, we establish that Algorithm 1 converges exactly to an integral local maximizer in a finite number of total iterations.

*Remark* 4.14. Building on the previous analyses, we next analyze the upper bound on the total number of iterations. By Remark 4.11, to trigger a saddle escape, the standard Frank–Wolfe steps must reduce the Frank–Wolfe gap to $\epsilon_1 = \min\left\{\frac{\xi_y^2}{16(\lambda+1)^2\tau^2 n^2}, \theta\right\}$. By (38), each iteration guarantees an ascent of at least $\min\left\{\frac{\epsilon_1^2}{4kL}, \frac{\epsilon_1}{2}\right\}$, and the maximum objective capacity is $O(k^2)$, the total number of standard Frank–Wolfe iterations before the last saddle escape is at most $O\left(\max\left\{\frac{k^3 L}{\epsilon_1^2}, \frac{k^2}{\epsilon_1}\right\}\right)$. By Corollary 4.13, the number of saddle escapes is at most $O(k^2 n^2)$. After the last saddle escape, the algorithm performs a final sequence of standard Frank–Wolfe steps to enter the basin of attraction of a local maximizer, which requires reducing the Frank–Wolfe gap to $\epsilon_2 = \min\left\{\frac{\sigma_x^2}{16\tau^2 L^2}, \theta\right\}$ by Theorem 4.1. The number of iterations for this is at most $O\left(\max\left\{\frac{k^3 L}{\epsilon_2^2}, \frac{k^2}{\epsilon_2}\right\}\right)$. Therefore, the total number of iterations is at most $O\left(\max\left\{\frac{k^3 L}{\epsilon_1^2}, \frac{k^2}{\epsilon_1}\right\} + k^2 n^2 + \max\left\{\frac{k^3 L}{\epsilon_2^2}, \frac{k^2}{\epsilon_2}\right\}\right)$.

## 5. Numerical Experiments

In this section, we first provide empirical evidence to corroborate our theoretical analysis. Our first two experiments

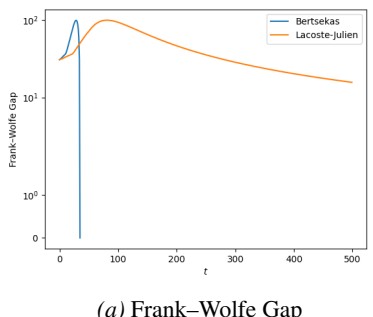 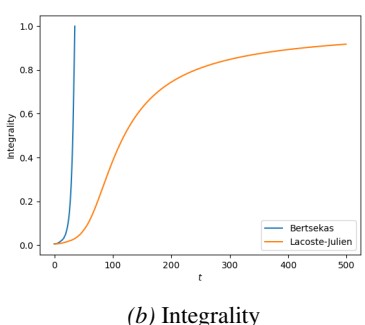 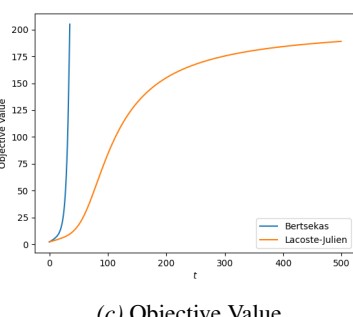

*(a)* Frank–Wolfe Gap        *(b)* Integrality        *(c)* Objective Value

*Figure 2.* **Finite-Step Exact Convergence on the Facebook Dataset.** Comparison between our step-size rule (blue) and the step-size rule proposed by Lacoste-Julien (2016) (orange). **(a):** The Frank–Wolfe gap is defined as $G(\boldsymbol{x}) := \max_{\boldsymbol{s} \in \mathcal{C}_k^n} \langle \nabla g(\boldsymbol{x}), \boldsymbol{s} - \boldsymbol{x} \rangle$, where $\mathcal{C}_k^n = \{\boldsymbol{y} \in [0,1]^n \mid \sum_{i \in [n]} y_i = k\}$. This metric characterizes the proximity to stationarity, vanishing if and only if $\boldsymbol{x}$ is a stationary point. **(b):** The integrality is defined as $\|\boldsymbol{x}\|_2^2 / k$. This metric characterizes the proximity to integrality, attaining one if and only if $\boldsymbol{x}$ is an integral point. **(c):** The objective value of $g(\boldsymbol{x})$.

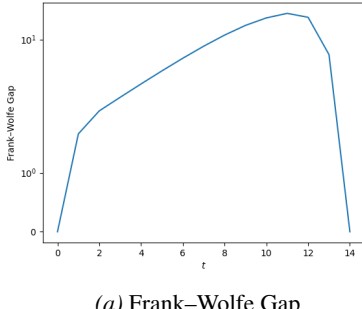 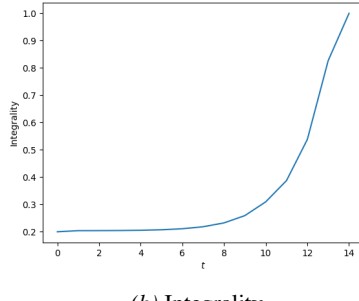 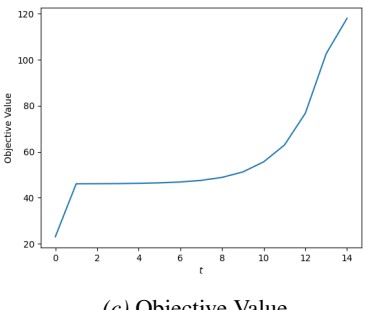

*(a)* Frank–Wolfe Gap        *(b)* Integrality        *(c)* Objective Value

*Figure 3.* **Saddle Escape on a Synthetic Regular Graph.**

are explicitly designed to verify two critical properties of the proposed algorithm: finite-step exact convergence and saddle escape capability. Furthermore, we compare SE-FW with the standard Frank–Wolfe algorithm on real-world datasets, and with greedy peeling on planted graphs, to demonstrate the effectiveness of our proposed algorithm. In our experiments, we set the penalty parameter $\lambda = 1.5$ and the initialization point $\boldsymbol{x}^{(0)} = \frac{k}{n} \mathbf{1}$ for SE-FW.

### 5.1. Verification of Finite-Step Convergence

We evaluate SE-FW on the SNAP Facebook dataset ($n = 4{,}039$, $m = 88{,}234$) (Leskovec & Krevl, 2014) with a subgraph size of $k = 20$. We compare our step-size rule $\gamma^{(t)} = \min\left\{1, \frac{(\boldsymbol{v}^{(t)})^\mathsf{T} \boldsymbol{d}^{(t)}}{L \|\boldsymbol{d}^{(t)}\|_2^2}\right\}$ (Bertsekas, 2016, p. 268) with another adaptive step-size rule $\gamma^{(t)} = \min\left\{1, \frac{(\boldsymbol{v}^{(t)})^\mathsf{T} \boldsymbol{d}^{(t)}}{2kL}\right\}$ (Lacoste-Julien, 2016).

Figure 2 corroborates our theoretical analysis. The baseline (orange) exhibits the expected sublinear asymptotic convergence, while our SE-FW (blue) demonstrates a faster convergence followed by a "final jump", where the Frank–

Wolfe gap drops to zero and the integrality attains one. This behavior empirically validates Theorem 4.1 and confirms the prediction in Remark 4.2 that the adaptive step-size rule proposed by Lacoste-Julien (2016) is too conservative to achieve finite-step exact convergence.

### 5.2. Verification of Saddle Escape Capability

Since every point that is not a local maximizer has a strictly ascent direction, standard Frank–Wolfe rarely gets stuck in the neighborhood of a saddle point in practice (refer to Subsection 5.3 for further details). To verify the saddle escape capability, we employ a synthetic dataset: a random 10-regular graph with 100 vertices generated by the Python package NetworkX (Steger & Wormald, 1999; Kim & Vu, 2003) and the subgraph size $k = 20$.

For a regular graph, the uniform initialization is a stationary point (as shown in Figure 3(a), where the Frank–Wolfe gap is exactly zero). Consequently, the standard Frank–Wolfe algorithm terminates at this point. With our saddle escape mechanism, we observe a sharp ascent at the beginning in Figure 3(c), and the algorithm eventually converges exactly

to an integral local maximizer.

### 5.3. Comparison with Standard Frank–Wolfe on Real-World Datasets

To demonstrate that the saddle escape mechanism is rarely triggered on real-world datasets due to the benign landscape, we compared the performance of SE-FW and standard Frank–Wolfe on the Facebook ($n = 4,039$, $m = 88,234$) and Web-Stanford ($n = 281,903$, $m = 1,992,636$) SNAP datasets (Leskovec & Krevl, 2014).

We set the penalty parameter $\lambda = 1$ and the initialization point $\boldsymbol{x}^{(0)} = \frac{k}{n}\mathbf{1}$ for standard Frank–Wolfe, following the settings in Lu et al. (2025). Table 1 demonstrates that the two algorithms exhibit nearly identical performance on these two real-world datasets.

*Table 1.* Comparison between Standard Fran–Wolfe and SE-FW on SNAP Datasets

| Dataset | $k$ | Algorithm | Iterations | Density |
|---|---|---|---|---|
| Facebook | 20 | Standard FW | 35 | 1.0 |
| | | SE-FW | 36 | 1.0 |
| | 50 | Standard FW | 15 | 1.0 |
| | | SE-FW | 15 | 1.0 |
| web-Stanford | 20 | Standard FW | 110 | 0.9895 |
| | | SE-FW | 110 | 0.9895 |
| | 50 | Standard FW | 67 | 0.7298 |
| | | SE-FW | 66 | 0.7298 |

### 5.4. Comparison with Greedy Peeling on Synthetic Planted Clique Graphs

To further demonstrate the effectiveness of our proposed algorithm, we compare the performance of SE-FW and greedy peeling (Asahiro et al., 2000; Charikar, 2000) on synthetic planted clique graphs.

We generated 20 Erdős–Rényi random graphs ($n = 10,000$, $p = 0.05$) each containing a planted clique of size $k = 30$, using consecutive random seeds from 0 to 19. Both greedy peeling and SE-FW were evaluated across these 20 instances. The results show that greedy peeling recovered the planted clique in only 2 out of 20 instances, whereas SE-FW succeeded in 9 instances. Notably, SE-FW succeeded in all cases where greedy peeling was successful. These results further demonstrate the effectiveness of our proposed framework.

## 6. Related Work

**Saddle Escape**: Escaping saddle points has been extensively studied in non-convex optimization. For uncon-

strained problems, Lee et al. (2016) applied the stable manifold theorem to prove that gradient descent almost surely converges to a local maximizer. Ge et al. (2015); Jin et al. (2017) proposed random perturbation-based approaches to escape saddle points. The framework proposed by Ge et al. (2015) can also be extended to handle equality-constrained problems. Note that our relaxed D$k$S problem includes inequality constraints. To address more general constrained problems, Mokhtari et al. (2018) proposed a framework for general convex sets. However, applying their framework to our relaxed D$k$S problem requires repeatedly approximating the solution of a generally non-convex quadratic programming problem, which is hard to do. In addition, Lu et al. (2020); Nouiehed & Razaviyayn (2020) also introduced perturbation-based algorithms for linearly constrained problems. Our proposed SE-FW leverages explicit positive curvature to escape saddle points and exactly converges to a local maximizer within *finite steps* rather than the asymptotic convergence achieved by all the aforementioned works.

## 7. Conclusion

In this work, we provide a comprehensive theoretical analysis of the diagonal loading-based non-convex relaxation for the Densest $k$-Subgraph problem, effectively bridging the gap between its empirical success and theoretical understanding. We establish the tightness of the relaxation and reveal a strict dichotomy in the optimization landscape: all integral stationary points are local maximizers, while all non-integral ones are strict saddle points with explicit positive curvature. Leveraging this benign geometry, we propose the Saddle-Escaping Frank–Wolfe algorithm, which guarantees a finite-step exact convergence to a local maximizer. Our experimental results further verify our theoretical findings.

### Acknowledgments

Qiheng Lu and Nicholas D. Sidiropoulos were partially supported by the Louis T. Rader Chair funds at UVA. Aritra Konar was supported by the KU Leuven Special Research Fund (BOF/STG-22-040).

## Impact Statement

This paper presents work whose goal is to advance the field of machine learning. There are many potential societal consequences of our work, none of which we feel must be specifically highlighted here.

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

## A. Proof of Theorem 2.3

*Proof.* Suppose that $x$ is a non-integral feasible point of (3). Let $\mathcal{M}(x) = \{i \in [n] \mid 0 < x_i < 1\}$. The set $\mathcal{M}(x)$ denotes the indices of all entries in $x$ that are not integral. Since the cardinality of $\mathcal{M}(x)$ is either 0 or strictly greater than 1, for any non-integral feasible $x$, we can always find two distinct vertices $i, j \in \mathcal{M}(x)$ such that $\lambda x_i + s_i \geq \lambda x_j + s_j$, where $s_i = \sum_{(i,l) \in \mathcal{E}} x_l, \forall i \in [n]$.

Let $\delta = \min\{x_j, 1 - x_i\}$ and $d = e_i - e_j$, where $e_i$ is the $i$-th vector of the canonical basis for $\mathbb{R}^n$, and $\hat{x} = x + \delta d$. Clearly, $\hat{x}$ is still feasible. Next, we consider the difference between $g(\hat{x})$ and $g(x)$ when $\lambda \geq 1$:

$$
\begin{aligned}
&g(\hat{x}) - g(x) \\
=& (x_i + \delta)(s_i - a_{ij}x_j) + \frac{\lambda}{2}(x_i + \delta)^2 + (x_j - \delta)(s_j - a_{ij}x_i) + \frac{\lambda}{2}(x_j - \delta)^2 + a_{ij}(x_i + \delta)(x_j - \delta) \\
&- x_i(s_i - a_{ij}x_j) - \frac{\lambda}{2}x_i^2 - x_j(s_j - a_{ij}x_i) - \frac{\lambda}{2}x_j^2 - a_{ij}x_ix_j \\
=& \delta(\lambda x_i + s_i - \lambda x_j - s_j) + (\lambda - a_{ij})\delta^2 \\
\geq& 0.
\end{aligned}
\tag{7}
$$

Hence, the objective function value $g(\hat{x})$ is greater than or equal to the objective function value $g(x)$ and the cardinality of $\mathcal{M}(\hat{x})$ is strictly smaller than the cardinality of $\mathcal{M}(x)$. Repeat this update until the cardinality of $\mathcal{M}(x)$ reduces to 0, and we obtain an integral feasible $x'$ of (3) such that $g(x') \geq g(x)$, which implies that there exists an integral feasible point that is optimal for (3). □

## B. Proof of Lemma 3.1

*Proof.* Suppose that $x$ is a non-integral feasible point of (3). Let $\mathcal{M}(x) = \{i \in [n] \mid 0 < x_i < 1\}$. The set $\mathcal{M}(x)$ denotes the indices of all entries in $x$ that are not integers. Since the cardinality of $\mathcal{M}(x)$ is either 0 or strictly greater than 1, for any non-integral feasible $x$, we can always find two distinct vertices $i, j \in \mathcal{M}(x)$ such that $\lambda x_i + s_i \geq \lambda x_j + s_j$, where $s_i = \sum_{(i,l) \in \mathcal{E}} x_l, \forall i \in [n]$.

Let $\hat{\delta} = \min\{x_j, 1 - x_i\}$ and $d = e_i - e_j$, where $e_i$ is the $i$-th vector of the canonical basis for $\mathbb{R}^n$. Clearly, $d$ is a feasible direction at $x$. Next, we consider the difference between $g(x + \delta d)$ and $g(x)$ when $\lambda > 1$ for every $\delta \in (0, \hat{\delta}]$:

$$
\begin{aligned}
&g(x + \delta d) - g(x) \\
=& (x_i + \delta)(s_i - a_{ij}x_j) + \frac{\lambda}{2}(x_i + \delta)^2 + (x_j - \delta)(s_j - a_{ij}x_i) + \frac{\lambda}{2}(x_j - \delta)^2 + a_{ij}(x_i + \delta)(x_j - \delta) \\
&- x_i(s_i - a_{ij}x_j) - \frac{\lambda}{2}x_i^2 - x_j(s_j - a_{ij}x_i) - \frac{\lambda}{2}x_j^2 - a_{ij}x_ix_j \\
=& \delta(\lambda x_i + s_i - \lambda x_j - s_j) + (\lambda - a_{ij})\delta^2 \\
>& 0.
\end{aligned}
\tag{8}
$$

Therefore, $d = e_i - e_j$ is a feasible strictly ascent direction at $x$, which implies that $x$ is not a local maximizer of (3). □

## C. Proof of Theorem 3.2

*Proof.* Since $x$ is a local maximizer of (3) with the penalty parameter $\lambda_1$, there exists $\epsilon > 0$, such that $x^\top(A + \lambda_1 I)x \geq y^\top(A + \lambda_1 I)y$ for every $y \in \mathcal{C}_\epsilon$, where $\mathcal{C}_\epsilon = \{y \in \mathcal{C}_k^n \mid \|x - y\|_2 \leq \epsilon\}$ and $\mathcal{C}_k^n = \{y \in [0,1]^n \mid \sum_{i \in [n]} y_i = k\}$. Now, we only need to show that $x^\top(A + \lambda_2 I)x \geq y^\top(A + \lambda_2 I)y$ for every $y \in \mathcal{C}_\epsilon$. From Lemma 3.1, we know that $x$ is

integral, which implies that

$$
\begin{aligned}
&\boldsymbol{x}^\mathsf{T}(\boldsymbol{A} + \lambda_2 \boldsymbol{I})\boldsymbol{x} - \boldsymbol{y}^\mathsf{T}(\boldsymbol{A} + \lambda_2 \boldsymbol{I})\boldsymbol{y} \\
=&\boldsymbol{x}^\mathsf{T}(\boldsymbol{A} + \lambda_1 \boldsymbol{I})\boldsymbol{x} + (\lambda_2 - \lambda_1)\|\boldsymbol{x}\|_2^2 - \boldsymbol{y}^\mathsf{T}(\boldsymbol{A} + \lambda_1 \boldsymbol{I})\boldsymbol{y} \\
&- (\lambda_2 - \lambda_1)\|\boldsymbol{y}\|_2^2 \\
\geq&(\lambda_2 - \lambda_1)(\|\boldsymbol{x}\|_2^2 - \|\boldsymbol{y}\|_2^2) \\
\geq&0,
\end{aligned}
\tag{9}
$$

where the last inequality holds because $\|\boldsymbol{z}\|_2^2$ is maximized over the feasible set of (3) when $\boldsymbol{z}$ is integral.

Therefore, $\boldsymbol{x}$ is also a local maximizer of (3) with the penalty parameter $\lambda_2$. $\qquad\square$

## D. Proof of Theorem 3.4

*Proof.* The Lagrangian function of (3) can be expressed as

$$
\begin{aligned}
L(\boldsymbol{x}, \mu, \boldsymbol{\alpha}, \boldsymbol{\beta}) =&g(\boldsymbol{x}) - \mu\left(\sum_{i\in[n]} x_i - k\right) \\
&+ \sum_{i\in[n]} \alpha_i x_i - \sum_{i\in[n]} \beta_i(x_i - 1).
\end{aligned}
\tag{10}
$$

The KKT conditions are

$$
\begin{cases}
v_i = \mu - \alpha_i + \beta_i, \\
\alpha_i \geq 0, \\
\beta_i \geq 0, \\
\alpha_i x_i = 0, \\
\beta_i(x_i - 1) = 0,
\end{cases}
\tag{11}
$$

for every $i \in [n]$, which is equivalent to $v_i \leq \mu$ for every $i \in \mathcal{S}_0$, $v_i \geq \mu$ for every $i \in \mathcal{S}_1$, and $v_i = \mu$ for every $i \in \mathcal{S}_f$. $\quad\square$

## E. Proof of Theorem 3.6

*Proof.* ($\Rightarrow$) First, we prove that if $\boldsymbol{x}$ is a local maximizer of (3), then $\boldsymbol{x}$ is integral and $\max_{i\in\mathcal{S}_0} v_i < \min_{i\in\mathcal{S}_1} v_i$.

By Lemma 3.1 and Corollary 3.5, we know that $\boldsymbol{x}$ is integral and $\max_{i\in\mathcal{S}_0} v_i \leq \min_{i\in\mathcal{S}_1} v_i$, so we only need to show that $\max_{i\in\mathcal{S}_0} v_i \neq \min_{i\in\mathcal{S}_1} v_i$.

Suppose that there exist $i \in \mathcal{S}_0$ and $j \in \mathcal{S}_1$ such that $v_i = v_j$. Let $\boldsymbol{d} = \boldsymbol{e}_i - \boldsymbol{e}_j$ where $\boldsymbol{e}_i$ is the $i$-th vector of the canonical basis for $\mathbb{R}^n$. For $\delta \in (0, 1]$, the exact second-order Taylor expansion at $\boldsymbol{x}$ is

$$
\begin{aligned}
g(\boldsymbol{x} + \delta\boldsymbol{d}) - g(\boldsymbol{x}) &= \delta\boldsymbol{v}^\mathsf{T}\boldsymbol{d} + \frac{1}{2}\delta^2 \boldsymbol{d}^\mathsf{T}(\boldsymbol{A} + \lambda\boldsymbol{I})\boldsymbol{d} \\
&= (v_i - v_j)\delta + (\lambda - a_{ij})\delta^2 \\
&> 0,
\end{aligned}
\tag{12}
$$

which contradicts the local optimality of $\boldsymbol{x}$.

Hence, $\boldsymbol{x}$ is a local maximizer of (3) when $\lambda > 1$ implies that $\boldsymbol{x}$ is integral and $\max_{i\in\mathcal{S}_0} v_i < \min_{i\in\mathcal{S}_1} v_i$.

($\Leftarrow$) Then, we prove that if $\boldsymbol{x}$ is integral and $\max_{i\in\mathcal{S}_0} v_i < \min_{i\in\mathcal{S}_1} v_i$, then $\boldsymbol{x}$ is a local maximizer of (3).

Since $\max_{i\in\mathcal{S}_0} v_i < \min_{i\in\mathcal{S}_1} v_i$, there exists a scalar $\mu$ such that $\max_{i\in\mathcal{S}_0} v_i < \mu < \min_{i\in\mathcal{S}_1} v_i$. Let $\sigma_i = |v_i - \mu|$ for every $i \in [n]$, and $\sigma_{\min} = \min_{i\in[n]} \sigma_i$.

Let a nonzero vector $\boldsymbol{d}$ be a direction for which $\boldsymbol{x} + \boldsymbol{d}$ remains feasible for (2). Since $\boldsymbol{x}$ is integral and $\boldsymbol{x} + \boldsymbol{d}$ is feasible for (2), we have $d_i \geq 0$ for every $i \in \mathcal{S}_0$, $d_i \leq 0$ for every $i \in \mathcal{S}_1$, and $\sum_{i\in[n]} d_i = 0$.

The exact second-order Taylor expansion at $x$ is

$$
\begin{aligned}
g(\boldsymbol{x} + \boldsymbol{d}) - g(\boldsymbol{x}) &= \boldsymbol{v}^{\mathsf{T}}\boldsymbol{d} + \frac{1}{2}\boldsymbol{d}^{\mathsf{T}}(\boldsymbol{A} + \lambda\boldsymbol{I})\boldsymbol{d} \\
&= \sum_{i \in [n]} v_i d_i + \frac{1}{2}\boldsymbol{d}^{\mathsf{T}}(\boldsymbol{A} + \lambda\boldsymbol{I})\boldsymbol{d}.
\end{aligned}
\tag{13}
$$

For the first-order term in (13), we have

$$
\begin{aligned}
\sum_{i \in [n]} v_i d_i &= \sum_{i \in [n]} (v_i - \mu)d_i \\
&= \sum_{i \in \mathcal{S}_0} (v_i - \mu)d_i + \sum_{i \in \mathcal{S}_1} (v_i - \mu)d_i \\
&= -\sum_{i \in \mathcal{S}_0} |v_i - \mu||d_i| - \sum_{i \in \mathcal{S}_1} |v_i - \mu||d_i| \\
&\leq -\sigma_{\min}\|\boldsymbol{d}\|_2
\end{aligned}
\tag{14}
$$

where the first equality is due to $\sum_{i \in [n]} d_i = 0$ and the last inequality is due to $\|\boldsymbol{d}\|_2 \leq \|\boldsymbol{d}\|_1$.

For the second-order term in (13), we have

$$
\boldsymbol{d}^{\mathsf{T}}(\boldsymbol{A} + \lambda\boldsymbol{I})\boldsymbol{d} \leq L\|\boldsymbol{d}\|_2^2.
\tag{15}
$$

Therefore, for every nonzero $\boldsymbol{d}$ such that $\|\boldsymbol{d}\|_2 < \frac{2\sigma_{\min}}{L}$, we have

$$
g(\boldsymbol{x} + \boldsymbol{d}) - g(\boldsymbol{x}) \leq -\sigma_{\min}\|\boldsymbol{d}\|_2 + \frac{L}{2}\|\boldsymbol{d}\|_2^2 < 0,
\tag{16}
$$

which implies that $x$ is a local maximizer. $\qquad\square$

## F. Proof of Theorem 3.10

*Proof.* Suppose that $x$ is a non-integral stationary point of (3). Since $x$ is non-integral, there exist two distinct indices $i, j \in \mathcal{S}_f$. Let $\boldsymbol{d} = \boldsymbol{e}_i - \boldsymbol{e}_j$ where $\boldsymbol{e}_i$ is the $i$-th vector of the canonical basis for $\mathbb{R}^n$. Since $i, j \in \mathcal{S}_f$, $\boldsymbol{d}$ is clearly a feasible direction at $x$.

Since $x$ is a stationary point, by Theorem 3.4, we have $v_i = v_j$, which implies that

$$
\boldsymbol{v}^{\mathsf{T}}\boldsymbol{d} = v_i - v_j = 0.
\tag{17}
$$

Since $\lambda > 1$, we also have

$$
\boldsymbol{d}^{\mathsf{T}}(\boldsymbol{A} + \lambda\boldsymbol{I})\boldsymbol{d} = 2(\lambda - a_{ij}) > 0.
\tag{18}
$$

Therefore, we can conclude that $x$ is a strict saddle point. $\qquad\square$

## G. Proof of Theorem 4.1

### G.1. Supporting Lemma

**Lemma G.1.** *Let $\boldsymbol{x}$ be a local maximizer of (3) with a non-integral $\lambda > 1$, and let $\sigma_{\boldsymbol{x}} = \min_{i \in \mathcal{S}_1(\boldsymbol{x})} v_i(\boldsymbol{x}) - \max_{i \in \mathcal{S}_0(\boldsymbol{x})} v_i(\boldsymbol{x}) > 0$ denote the gradient gap at $\boldsymbol{x}$. For any feasible point $\boldsymbol{y}$ of (3), if $\|\boldsymbol{x} - \boldsymbol{y}\|_2 < \frac{\sigma_{\boldsymbol{x}}}{2L}$, then $\boldsymbol{x}$ is the unique solution to the LMO (4) with respect to the gradient $\boldsymbol{v}(\boldsymbol{y})$.*

*Proof.* By the norm inequality and the $L$-smoothness condition, we have

$$
\|\boldsymbol{v}(\boldsymbol{x}) - \boldsymbol{v}(\boldsymbol{y})\|_\infty \leq \|\boldsymbol{v}(\boldsymbol{x}) - \boldsymbol{v}(\boldsymbol{y})\|_2 \leq L\|\boldsymbol{x} - \boldsymbol{y}\|_2 < \frac{\sigma_{\boldsymbol{x}}}{2},
\tag{19}
$$

which implies that

$$\min_{i \in \mathcal{S}_1(\boldsymbol{x})} v_i(\boldsymbol{y}) > \min_{i \in \mathcal{S}_1(\boldsymbol{x})} v_i(\boldsymbol{x}) - \frac{\sigma_{\boldsymbol{x}}}{2},$$
$$\max_{i \in \mathcal{S}_0(\boldsymbol{x})} v_i(\boldsymbol{y}) < \max_{i \in \mathcal{S}_0(\boldsymbol{x})} v_i(\boldsymbol{x}) + \frac{\sigma_{\boldsymbol{x}}}{2}. \tag{20}$$

Since $\min_{i \in \mathcal{S}_1(\boldsymbol{x})} v_i(\boldsymbol{x}) - \max_{i \in \mathcal{S}_0(\boldsymbol{x})} v_i(\boldsymbol{x}) = \sigma_{\boldsymbol{x}}$, we have

$$\min_{i \in \mathcal{S}_1(\boldsymbol{x})} v_i(\boldsymbol{y}) > \max_{i \in \mathcal{S}_0(\boldsymbol{x})} v_i(\boldsymbol{y}), \tag{21}$$

which implies that $\mathcal{S}_1(\boldsymbol{x})$ is the unique set of indices of the largest $k$ values of $\boldsymbol{v}(\boldsymbol{y})$. $\qquad\square$

### G.2. Proof of Theorem 4.1

*Proof.* Since $\|\boldsymbol{x}^{(t)} - \boldsymbol{x}\|_2 \leq \frac{\sigma_{\boldsymbol{x}}}{4L} < \frac{\sigma_{\boldsymbol{x}}}{2L}$, by Lemma G.1, we have $\boldsymbol{s}^{(t)} = \boldsymbol{x}$, which implies that $\boldsymbol{d}^{(t)} = \boldsymbol{x} - \boldsymbol{x}^{(t)}$. Hence, we only need to show that $(\boldsymbol{v}(\boldsymbol{x}^{(t)}))^{\mathsf{T}} \boldsymbol{d}^{(t)} \geq L \|\boldsymbol{d}^{(t)}\|_2^2$. For the left-hand side (LHS), we have

$$\begin{aligned} (\boldsymbol{v}(\boldsymbol{x}^{(t)}))^{\mathsf{T}} \boldsymbol{d}^{(t)} &= (\boldsymbol{x}^{(t)})^{\mathsf{T}} (\boldsymbol{A} + \lambda \boldsymbol{I}) \boldsymbol{d}^{(t)} \\ &= (\boldsymbol{x} - \boldsymbol{d}^{(t)})^{\mathsf{T}} (\boldsymbol{A} + \lambda \boldsymbol{I}) \boldsymbol{d}^{(t)} \\ &= \boldsymbol{x}^{\mathsf{T}} (\boldsymbol{A} + \lambda \boldsymbol{I}) \boldsymbol{d}^{(t)} - (\boldsymbol{d}^{(t)})^{\mathsf{T}} (\boldsymbol{A} + \lambda \boldsymbol{I}) \boldsymbol{d}^{(t)} \\ &\geq (\boldsymbol{v}(\boldsymbol{x}))^{\mathsf{T}} \boldsymbol{d}^{(t)} - L \|\boldsymbol{d}^{(t)}\|_2^2, \end{aligned} \tag{22}$$

which implies that we only need to show that $(\boldsymbol{v}(\boldsymbol{x}))^{\mathsf{T}} \boldsymbol{d}^{(t)} \geq 2L \|\boldsymbol{d}^{(t)}\|_2^2$.

Since $\boldsymbol{x}$ is integral, we have $d_i^{(t)} \leq 0$ for every $i \in \mathcal{S}_0(\boldsymbol{x})$ and $d_i^{(t)} \geq 0$ for every $i \in \mathcal{S}_1(\boldsymbol{x})$. Let $\rho = \max_{i \in \mathcal{S}_0(\boldsymbol{x})} v_i(\boldsymbol{x}) + \frac{\sigma_{\boldsymbol{x}}}{2} = \min_{i \in \mathcal{S}_1(\boldsymbol{x})} v_i(\boldsymbol{x}) - \frac{\sigma_{\boldsymbol{x}}}{2}$, we have

$$\begin{aligned} (\boldsymbol{v}(\boldsymbol{x}))^{\mathsf{T}} \boldsymbol{d}^{(t)} &= \sum_{i \in [n]} (v_i(\boldsymbol{x}) - \rho) d_i^{(t)} \\ &= \sum_{i \in \mathcal{S}_0(\boldsymbol{x})} (v_i(\boldsymbol{x}) - \rho) d_i^{(t)} + \sum_{i \in \mathcal{S}_1(\boldsymbol{x})} (v_i(\boldsymbol{x}) - \rho) d_i^{(t)} \\ &= \sum_{i \in \mathcal{S}_0(\boldsymbol{x})} |v_i(\boldsymbol{x}) - \rho| |d_i^{(t)}| + \sum_{i \in \mathcal{S}_1(\boldsymbol{x})} |v_i(\boldsymbol{x}) - \rho| |d_i^{(t)}| \\ &\geq \frac{\sigma_{\boldsymbol{x}}}{2} \|\boldsymbol{d}^{(t)}\|_2, \end{aligned} \tag{23}$$

where the first equality is due to $\sum_{i \in [n]} d_i^{(t)} = 0$ and the last inequality is due to $\|\boldsymbol{d}^{(t)}\|_1 \geq \|\boldsymbol{d}^{(t)}\|_2$.

Finally, because $\|\boldsymbol{d}^{(t)}\|_2 = \|\boldsymbol{x}^{(t)} - \boldsymbol{x}\|_2 \leq \frac{\sigma_{\boldsymbol{x}}}{4L}$, we have $(\boldsymbol{v}(\boldsymbol{x}))^{\mathsf{T}} \boldsymbol{d}^{(t)} \geq \frac{\sigma_{\boldsymbol{x}}}{2} \|\boldsymbol{d}^{(t)}\|_2 \geq 2L \|\boldsymbol{d}^{(t)}\|_2^2$. $\qquad\square$

## H. Proof of Theorem 4.5

### H.1. Supporting Lemma

**Lemma H.1.** *Let $\boldsymbol{x}$ be a non-integral feasible point of* (3), *and let $\boldsymbol{y}$ be any integral feasible point of* (3) *closest to $\boldsymbol{x}$ (in the Euclidean norm). Then the following properties hold:*

1. *$y_i = x_i$ for every $i \in \mathcal{S}_0(\boldsymbol{x}) \cup \mathcal{S}_1(\boldsymbol{x})$.*

2. *For any $i \in \mathcal{S}_1(\boldsymbol{y})$ and $j \in \mathcal{S}_0(\boldsymbol{y})$, we have $x_i \geq x_j$.*

*Proof.* We first consider the first property. Suppose that there exists $i \in \mathcal{S}_0(\boldsymbol{x})$ such that $y_i = 1$. Since $\sum_{i \in [n]} x_i = \sum_{i \in [n]} y_i = k$, we have $\sum_{j \neq i} (x_j - y_j) = 1$, which implies that there exists $j \neq i$ such that $y_j = 0$ and $x_j > 0$. Let $z_i = 0$,

$z_j = 1$, and $z_l = y_l$ for every $l \notin \{i, j\}$, then we have $\|\boldsymbol{y} - \boldsymbol{x}\|_2^2 - \|\boldsymbol{z} - \boldsymbol{x}\|_2^2 = 2x_j > 0$, which contradicts the assumption that $\boldsymbol{y}$ is the integral feasible point closest to $\boldsymbol{x}$.

Similarly, suppose that there exists $i \in \mathcal{S}_1(\boldsymbol{x})$ such that $y_i = 0$. Since $\sum_{i \in [n]} x_i = \sum_{i \in [n]} y_i = k$, we have $\sum_{j \neq i} (y_j - x_j) = 1$ which implies that there exists $j \neq i$ such that $y_j = 1$ and $x_j < 1$. Let $z_i = 1$, $z_j = 0$, and $z_l = y_l$ for every $l \notin \{i, j\}$, then we have $\|\boldsymbol{y} - \boldsymbol{x}\|_2^2 - \|\boldsymbol{z} - \boldsymbol{x}\|_2^2 = 2(1 - x_j) > 0$, which contradicts the assumption that $\boldsymbol{y}$ is the integral feasible point closest to $\boldsymbol{x}$.

Therefore, we can conclude that $y_i = x_i$ for every $i \in \mathcal{S}_0(\boldsymbol{x}) \cup \mathcal{S}_1(\boldsymbol{x})$.

Next, we consider the second property. Suppose that there exist $i \in \mathcal{S}_1(\boldsymbol{y})$ and $j \in \mathcal{S}_0(\boldsymbol{y})$ such that $x_i < x_j$. Let $z_i = 0$, $z_j = 1$, and $z_l = y_l$ for every $l \notin \{i, j\}$, then we have $\|\boldsymbol{y} - \boldsymbol{x}\|_2^2 - \|\boldsymbol{z} - \boldsymbol{x}\|_2^2 = 2(x_j - x_i) > 0$, which contradicts the assumption that $\boldsymbol{y}$ is the integral feasible point closest to $\boldsymbol{x}$. $\qquad\square$

## H.2. Proof of Theorem 4.5

*Proof.* By the first property in Lemma H.1, we know that $1 \leq \sum_{i \in \mathcal{S}_f(\boldsymbol{x})} x_i = \sum_{i \in \mathcal{S}_f(\boldsymbol{x})} y_i < |\mathcal{S}_f(\boldsymbol{x})|$, which implies that $\mathcal{S}_1(\boldsymbol{y}) \cap \mathcal{S}_f(\boldsymbol{x})$ and $\mathcal{S}_0(\boldsymbol{y}) \cap \mathcal{S}_f(\boldsymbol{x})$ are non-empty sets because of the pigeonhole principle. Let $j \in \arg\min_{i \in \mathcal{S}_1(\boldsymbol{y}) \cap \mathcal{S}_f(\boldsymbol{x})} x_i$ and $l \in \arg\max_{i \in \mathcal{S}_0(\boldsymbol{y}) \cap \mathcal{S}_f(\boldsymbol{x})} x_i$. By the second property in Lemma H.1, we know that $x_j \geq x_l$, so we only need to derive a lower bound for $\min\{x_l, 1 - x_j\}$.

Since $\sum_{i \in [n]} x_i = \sum_{i \in [n]} y_i = k$, we have

$$M = \sum_{i \in \mathcal{S}_1(\boldsymbol{y}) \cap \mathcal{S}_f(\boldsymbol{x})} (1 - x_i) = \sum_{i \in \mathcal{S}_0(\boldsymbol{y}) \cap \mathcal{S}_f(\boldsymbol{x})} x_i, \tag{24}$$

where $M$ is a positive scalar. By the definitions of $x_j$ and $x_l$, we have

$$
\begin{aligned}
1 - x_j &\geq \frac{M}{|\mathcal{S}_1(\boldsymbol{y}) \cap \mathcal{S}_f(\boldsymbol{x})|} \geq \frac{M}{n}, \\
x_l &\geq \frac{M}{|\mathcal{S}_0(\boldsymbol{y}) \cap \mathcal{S}_f(\boldsymbol{x})|} \geq \frac{M}{n}.
\end{aligned}
\tag{25}
$$

Hence, we only need to derive a lower bound for $M$. Next, we will consider the following two cases separately to derive this lower bound: when $\boldsymbol{y}$ is a stationary point and when it is not.

If $\boldsymbol{y}$ is a stationary point (which implies that $\boldsymbol{y}$ is an integral local maximizer by Theorem 3.7), then $\xi_{\boldsymbol{y}} = \min_{i \in \mathcal{S}_1(\boldsymbol{y})} v_i(\boldsymbol{y}) - \max_{i \in \mathcal{S}_0(\boldsymbol{y})} v_i(\boldsymbol{y}) > 0$. Let $\boldsymbol{z} = \boldsymbol{y} - \boldsymbol{x}$, by Theorem 3.4 and the definition of $M$, we have

$$
\begin{aligned}
\xi_{\boldsymbol{y}} &\leq v_j(\boldsymbol{y}) - v_l(\boldsymbol{y}) \\
&= v_j(\boldsymbol{y}) - v_l(\boldsymbol{y}) + v_l(\boldsymbol{x}) - v_j(\boldsymbol{x}) \\
&= [v_j(\boldsymbol{y}) - v_j(\boldsymbol{x})] - [v_l(\boldsymbol{y}) - v_l(\boldsymbol{x})] \\
&= (\boldsymbol{e}_j - \boldsymbol{e}_l)^\mathsf{T} (\boldsymbol{A} + \lambda \boldsymbol{I}) \boldsymbol{z} \\
&= \lambda(z_j - z_l) + \sum_{i \in \mathcal{S}_1(\boldsymbol{y}) \cap \mathcal{S}_f(\boldsymbol{x})} (a_{ji} - a_{li}) z_i + \sum_{i \in \mathcal{S}_0(\boldsymbol{y}) \cap \mathcal{S}_f(\boldsymbol{x})} (a_{ji} - a_{li}) z_i \\
&\leq \lambda((1 - x_j) + x_l) + \sum_{i \in \mathcal{S}_1(\boldsymbol{y}) \cap \mathcal{S}_f(\boldsymbol{x})} |a_{ji} - a_{li}| |1 - x_i| + \sum_{i \in \mathcal{S}_0(\boldsymbol{y}) \cap \mathcal{S}_f(\boldsymbol{x})} |a_{ji} - a_{li}| | - x_i| \\
&\leq 2\lambda M + \sum_{i \in \mathcal{S}_1(\boldsymbol{y}) \cap \mathcal{S}_f(\boldsymbol{x})} (1 - x_i) + \sum_{i \in \mathcal{S}_0(\boldsymbol{y}) \cap \mathcal{S}_f(\boldsymbol{x})} x_i \\
&= 2(\lambda + 1) M,
\end{aligned}
\tag{26}
$$

which implies that $M \geq \frac{\xi_{\boldsymbol{y}}}{2(\lambda+1)}$.

If $\boldsymbol{y}$ is not a stationary point, then there exist $p \in \mathcal{S}_0(\boldsymbol{y})$ and $q \in \mathcal{S}_1(\boldsymbol{y})$ such that $\xi_{\boldsymbol{y}} = v_p(\boldsymbol{y}) - v_q(\boldsymbol{y}) > 0$. By the second property in Lemma H.1, we have $x_p \leq x_q$. Specifically, this covers three possible scenarios: $0 < x_p \leq x_q < 1$,

$x_p < x_q = 1$, and $x_q > x_p = 0$. In all such cases, it follows from Theorem 3.4 that $v_p(\boldsymbol{x}) \le v_q(\boldsymbol{x})$. Let $\boldsymbol{z} = \boldsymbol{y} - \boldsymbol{x}$, by the definition of $M$, we have

$$
\begin{aligned}
\xi_{\boldsymbol{y}} &= v_p(\boldsymbol{y}) - v_q(\boldsymbol{y}) \\
&\le v_p(\boldsymbol{y}) - v_q(\boldsymbol{y}) + v_q(\boldsymbol{x}) - v_p(\boldsymbol{x}) \\
&= [v_p(\boldsymbol{y}) - v_p(\boldsymbol{x})] - [v_q(\boldsymbol{y}) - v_q(\boldsymbol{x})] \\
&= (\boldsymbol{e}_p - \boldsymbol{e}_q)^{\mathsf{T}} (\boldsymbol{A} + \lambda \boldsymbol{I}) \boldsymbol{z} \\
&= \lambda(z_p - z_q) + \sum_{i \in \mathcal{S}_1(\boldsymbol{y}) \cap \mathcal{S}_f(\boldsymbol{x})} (a_{pi} - a_{qi}) z_i + \sum_{i \in \mathcal{S}_0(\boldsymbol{y}) \cap \mathcal{S}_f(\boldsymbol{x})} (a_{pi} - a_{qi}) z_i \\
&\le \lambda(-x_p - (1 - x_q)) + \sum_{i \in \mathcal{S}_1(\boldsymbol{y}) \cap \mathcal{S}_f(\boldsymbol{x})} |a_{pi} - a_{qi}| |1 - x_i| + \sum_{i \in \mathcal{S}_0(\boldsymbol{y}) \cap \mathcal{S}_f(\boldsymbol{x})} |a_{pi} - a_{qi}| |-x_i| \\
&\le \sum_{i \in \mathcal{S}_1(\boldsymbol{y}) \cap \mathcal{S}_f(\boldsymbol{x})} (1 - x_i) + \sum_{i \in \mathcal{S}_0(\boldsymbol{y}) \cap \mathcal{S}_f(\boldsymbol{x})} x_i \\
&= 2M,
\end{aligned}
\tag{27}
$$

which implies that $M \ge \frac{\xi_{\boldsymbol{y}}}{2}$.

Therefore, we can conclude that $\min\{x_l, 1 - x_j\} \ge \frac{\xi_{\boldsymbol{y}}}{2(\lambda + 1)n}$. $\qquad\square$

## I. Discussion on the Order-wise Tightness of Theorem 4.5

*Remark* I.1. When $1 < \lambda < 2$, we have the lower bound $\xi_{\boldsymbol{y}} \ge \min\{\lambda - 1, 2 - \lambda\}$. By choosing the penalty parameter $\lambda$ such that $\lambda - 1 = \Theta(1)$ and $2 - \lambda = \Theta(1)$ (e.g., $\lambda = 1.5$), $\min\{x_{(k+1)}, 1 - x_{(k)}\}$ is lower-bounded by $\Omega(1/n)$. The order of this bound cannot be improved in general. Consider a graph instance constructed by a clique of size $k - 1$ and an independent set of size $n - k + 1$, where every node in the $(k - 1)$-clique is fully connected to every node in the independent set. We can easily verify that $\boldsymbol{x} = [\underbrace{1, \ldots, 1}_{k-1}, \underbrace{\frac{1}{n-k+1}, \ldots, \frac{1}{n-k+1}}_{n-k+1}]^{\mathsf{T}}$ is a stationary point of this instance. Since

$\min\{x_i, 1 - x_i, x_j, 1 - x_j\} = O(1/n)$ for every pair of distinct indices $i$ and $j$ when $n - k = \Theta(n)$, we can confirm that the lower bound obtained in Theorem 4.5 is order-wise tight.

## J. Proof of Lemma 4.8

*Proof.* In order to prove $|x_{(i)} - y_{(i)}| \le \epsilon$, we only need to prove $y_{(i)} - \epsilon \le x_{(i)} \le y_{(i)} + \epsilon$ for every $i \in [n]$.

We first prove that $x_{(i)} \le y_{(i)} + \epsilon$ for every $i \in [n]$. Suppose that there exists $i \in [n]$ such that $x_{(i)} > y_{(i)} + \epsilon$. Let $\mathcal{S} = \{j \in [n] \mid x_j \ge x_{(i)}\}$, we have $|\mathcal{S}| \ge i$ and $x_j \ge x_{(i)} > y_{(i)} + \epsilon$ for every $j \in \mathcal{S}$. Since $\|\boldsymbol{x} - \boldsymbol{y}\|_\infty \le \epsilon$, we have $y_j + \epsilon \ge x_j \ge x_{(i)} > y_{(i)} + \epsilon$ for every $j \in \mathcal{S}$. Since $|\mathcal{S}| \ge i$, there exist at least $i$ elements in $\boldsymbol{y}$ that are strictly greater than $y_{(i)}$, which contradicts the definition of $y_{(i)}$.

Similarly, by symmetry, we can prove $y_{(i)} \le x_{(i)} + \epsilon$ for every $i \in [n]$.

Therefore, we can conclude that $|x_{(i)} - y_{(i)}| \le \epsilon$ for every $i \in [n]$. $\qquad\square$

## K. Proof of Theorem 4.10

*Proof.* Let $\mathcal{C}_k^n = \{\boldsymbol{x} \in [0, 1]^n \mid \sum_{i \in [n]} x_i = k\}$ denote the constraint set of problem (3), and define the Euclidean projection of a point $\boldsymbol{y}$ onto $\mathcal{C}_k^n$ as

$$
\Pi_{\mathcal{C}_k^n}(\boldsymbol{y}) = \arg\min_{\boldsymbol{x} \in \mathcal{C}_k^n} \frac{1}{2} \|\boldsymbol{x} - \boldsymbol{y}\|_2^2
\tag{28}
$$

Furthermore, let $R(\boldsymbol{x}) := \boldsymbol{x} - \Pi_{\mathcal{C}_k^n}(\boldsymbol{x} + \nabla g(\boldsymbol{x}))$ denote the residual of the projected gradient at $\boldsymbol{x} \in \mathcal{C}_k^n$.

We first prove that $\|R(\boldsymbol{x})\|_2 \le \sqrt{G(\boldsymbol{x})}$. By the property of Euclidean projection, we have

$$
\langle \boldsymbol{x} + \nabla g(\boldsymbol{x}) - \Pi_{\mathcal{C}_k^n}(\boldsymbol{x} + \nabla g(\boldsymbol{x})), \boldsymbol{z} - \Pi_{\mathcal{C}_k^n}(\boldsymbol{x} + \nabla g(\boldsymbol{x})) \rangle \le 0,
\tag{29}
$$

for every $\boldsymbol{z} \in \mathcal{C}_k^n$. Substituting $\boldsymbol{z} = \boldsymbol{x}$, we have

$$\langle \boldsymbol{x} + \nabla g(\boldsymbol{x}) - \Pi_{\mathcal{C}_k^n}(\boldsymbol{x} + \nabla g(\boldsymbol{x})), \boldsymbol{x} - \Pi_{\mathcal{C}_k^n}(\boldsymbol{x} + \nabla g(\boldsymbol{x})) \rangle \leq 0. \tag{30}$$

On re-arranging, we obtain

$$\|\boldsymbol{x} - \Pi_{\mathcal{C}_k^n}(\boldsymbol{x} + \nabla g(\boldsymbol{x}))\|_2^2 \leq \langle \nabla g(\boldsymbol{x}), \Pi_{\mathcal{C}_k^n}(\boldsymbol{x} + \nabla g(\boldsymbol{x})) - \boldsymbol{x} \rangle. \tag{31}$$

Observe that the LHS of the above inequality is precisely $\|R(\boldsymbol{x})\|_2^2$. Since the definition of Frank–Wolfe gap is $G(\boldsymbol{x}) = \max_{\boldsymbol{s} \in \mathcal{C}_k^n} \langle \nabla g(\boldsymbol{x}), \boldsymbol{s} - \boldsymbol{x} \rangle$, we have

$$\langle \nabla g(\boldsymbol{x}), \Pi_{\mathcal{C}_k^n}(\boldsymbol{x} + \nabla g(\boldsymbol{x})) - \boldsymbol{x} \rangle \leq G(\boldsymbol{x}). \tag{32}$$

Chaining the final pair of inequalities then yields the desired result $\|R(\boldsymbol{x})\|_2^2 \leq G(\boldsymbol{x}) \leq \theta$.

By Theorem 2.3 in Luo & Tseng (1992), we have

$$\|\boldsymbol{x} - \boldsymbol{y}\|_\infty \leq \|\boldsymbol{x} - \boldsymbol{y}\|_2 \leq \tau \|\boldsymbol{x} - \Pi_{\mathcal{C}_k^n}(\boldsymbol{x} + \nabla g(\boldsymbol{x}))\|_2 \leq \tau \sqrt{G(\boldsymbol{x})}. \tag{33}$$

$\square$

## L. Proof of Theorem 4.12

*Proof.* By the ascent lemma, we have

$$g(\boldsymbol{x}^{(t+1)}) = g(\boldsymbol{x}^{(t)} + \gamma^{(t)}\boldsymbol{d}^{(t)}) \geq g(\boldsymbol{x}^{(t)}) + \gamma^{(t)}(\boldsymbol{v}^{(t)})^\mathsf{T}\boldsymbol{d}^{(t)} - \frac{1}{2}(\gamma^{(t)})^2 L \|\boldsymbol{d}^{(t)}\|_2^2. \tag{34}$$

By the definition of the Frank–Wolfe gap, we have

$$g(\boldsymbol{x}^{(t+1)}) \geq g(\boldsymbol{x}^{(t)}) + \gamma^{(t)} G(\boldsymbol{x}^{(t)}) - \frac{1}{2}(\gamma^{(t)})^2 L \|\boldsymbol{d}^{(t)}\|_2^2, \tag{35}$$

where $G(\boldsymbol{x}^{(t)})$ is the Frank–Wolfe gap at $\boldsymbol{x}^{(t)}$.

If $\gamma^{(t)} = \frac{G(\boldsymbol{x}^{(t)})}{L\|\boldsymbol{d}^{(t)}\|_2^2}$, then we have

$$g(\boldsymbol{x}^{(t+1)}) - g(\boldsymbol{x}^{(t)}) \geq \gamma^{(t)} G(\boldsymbol{x}^{(t)}) - \frac{1}{2}(\gamma^{(t)})^2 L \|\boldsymbol{d}^{(t)}\|_2^2 = \frac{G(\boldsymbol{x}^{(t)})^2}{2L\|\boldsymbol{d}^{(t)}\|_2^2} \geq \frac{G(\boldsymbol{x}^{(t)})^2}{4kL}. \tag{36}$$

If $\gamma^{(t)} = 1$, which implies that $G(\boldsymbol{x}^{(t)}) \geq L\|\boldsymbol{d}^{(t)}\|_2^2$, then we have

$$g(\boldsymbol{x}^{(t+1)}) - g(\boldsymbol{x}^{(t)}) \geq G(\boldsymbol{x}^{(t)}) - \frac{1}{2}L\|\boldsymbol{d}^{(t)}\|_2^2 \geq \frac{G(\boldsymbol{x}^{(t)})}{2}. \tag{37}$$

Hence, we have

$$g(\boldsymbol{x}^{(t+1)}) - g(\boldsymbol{x}^{(t)}) \geq \min\left\{ \frac{G(\boldsymbol{x}^{(t)})^2}{4kL}, \frac{G(\boldsymbol{x}^{(t)})}{2} \right\}, \tag{38}$$

which implies that the objective values are strictly monotonically increasing until a stationary point is reached and

$$h_0 \geq g(\boldsymbol{x}^{(T+1)}) - g(\boldsymbol{x}^{(0)}) \geq (T+1)\min\left\{ \frac{\tilde{G}_T^2}{4kL}, \frac{\tilde{G}_T}{2} \right\}. \tag{39}$$

If $\tilde{G}_T > 2kL$, we have $h_0 \geq \frac{(T+1)\tilde{G}_T}{2}$, which implies that $\tilde{G}_T \leq \frac{2h_0}{T+1}$. If $\tilde{G}_T \leq 2kL$, we have $h_0 \geq \frac{(T+1)\tilde{G}_T^2}{4kL}$, which implies that $\tilde{G}_T \leq \sqrt{\frac{4h_0kL}{T+1}}$. $\square$

