# OpenReview forum: "On Densest $k$-Subgraph Mining and Diagonal Loading: Optimization Landscape and Finite-Step Exact Convergence Analysis"
_ICML.cc/2026/Conference — ICML 2026 regular_

### Official Review · Reviewer_bCTa · 2026-03-06

**Soundness:** 4
**Presentation:** 4
**Significance:** 3
**Originality:** 3
**Overall Recommendation:** 5
**Confidence:** 3

**Summary:**

The authors present a thorough theoretical analysis of the recently introduced penalty-based relaxation for the Densest k-subgraph problem. They establish key geometric properties of this relaxation which provide a strong theoretical basis for their new saddle-escaping algorithm, which they prove converges to an integral local maximizer in finite steps.

**Compliance With Llm Reviewing Policy:**

Affirmed.

**Key Questions For Authors:**

Are there any circumstances where a practitioner might prefer the simpler method without saddle escaping? Never?

Is a similar relaxation technique applicable to the max quasi clique problem? This seems to be more relevant to bioinformatics.

**Limitations:**

Yes

**Strengths And Weaknesses:**

The theoretical results are significant and the paper's presentation is very clear.

My main concern was the lack of comprehensive experiments. The graphs used are significantly smaller than the ones used by Lu et al. (2025) and there is no straight comparison to the Frank-Wolfe version without the saddle escaping machinery. Practitioners will be interested in the cost/performance trade-off when deciding if the saddle escaping machinery is actually worth it. Also it seems redundant to include the learning rate of Lacoste-Julien, given Lu et al. (2025) had already shown that it isn’t as effective as the learning rate of Bertsekas, (2016) in practice.

---

> ### Author Rebuttal · Authors · 2026-03-27
>
> Thank you for your review. Here are our responses:
>
> >My main concern was the lack of comprehensive experiments. The graphs used are significantly smaller than the ones used by Lu et al. (2025) and there is no straight comparison to the Frank-Wolfe version without the saddle escaping machinery. Practitioners will be interested in the cost/performance trade-off when deciding if the saddle escaping machinery is actually worth it.
>
> **Response:** We clarify that the primary goal of this paper is to provide a rigorous theoretical explanation for the empirical success of Frank–Wolfe in Lu et al. (2025). As discussed in Section 5.2, all non-integral stationary points are strict saddles with explicit positive curvature. It is our empirical observation from all real-world datasets used in Lu et al. (2025) that even the standard Frank–Wolfe with $\lambda = 1$ consistently escapes saddles and converges to an integral point in a finite number of steps. Therefore, the saddle-escaping mechanism is there for theoretical completeness and it is only triggered in "engineered" cases, such as the regular graph used in this paper. Comprehensive empirical comparisons against baselines on real-world datasets are already provided in Lu et al. (2025); repeating them here would not yield additional insight. The two experiments in this paper are specifically designed to verify the two key theoretical results: finite-step exact convergence and saddle escaping capability. We will include these clarifications in the final version.
>
> >Also it seems redundant to include the learning rate of Lacoste-Julien, given Lu et al. (2025) had already shown that it isn’t as effective as the learning rate of Bertsekas, (2016) in practice.
>
> **Response:** While Lu et al. (2025) indeed observed that the step-size rule of Lacoste-Julien (2016) converges more slowly in practice, we include this comparison to clearly illustrate the theoretical analysis in Remark 4.2, which explains why this step-size rule prevents finite-step exact convergence due to its vanishing step size.
>
> >Is a similar relaxation technique applicable to the max quasi clique problem? This seems to be more relevant to bioinformatics.
>
> **Response:** Directly applying this framework to the max quasi-clique (MQC) problem, which maximizes subgraph size subject to a density constraint, is mathematically non-trivial. However, using our principled and scalable framework, one can iterate through different $k$ from $n$ down to 1 until finding a subgraph that satisfies the density requirement. We also note that a related connection between D$k$S and the optimal quasi-clique (OQC) problem has been established in Konar & Sidiropoulos (2024), which suggests that our framework may have broader applicability in quasi-clique related problems.
>
> A. Konar and N. D. Sidiropoulos. Optimal Quasi-clique: Hardness, Equivalence with Densest-k-Subgraph, and Quasi-partitioned Community Mining. AAAI 2024.

---

> > ### Author Rebuttal · Reviewer_bCTa · 2026-04-01
> >
> > Thanks. My concerns have been adequately addressed.

---

### Official Review · Reviewer_Y2Jg · 2026-03-10

**Soundness:** 3
**Presentation:** 3
**Significance:** 3
**Originality:** 3
**Overall Recommendation:** 4
**Confidence:** 3

**Summary:**

This paper presents a theoretical analysis of the recent diagonal-loading relaxation for Densest k-Subgraph. It proves tightness for λ≥1, characterizes the stationary landscape when λ>1 (integral stationary points are local maximizers and non-integral ones are strict saddles), and proposes a saddle-escaping Frank–Wolfe variant with finite-step exact convergence to an integral local maximizer.

**Compliance With Llm Reviewing Policy:**

Affirmed.

**Final Justification:**

My overall assessment of the paper is positive. I had some questions which the authors adequately answered. The main reason I am keeping my score is because I am not an expert on this topic and it is a bit hard for me to gauge impact here. Otherwise, the results seem important, and the paper is well written.

**Key Questions For Authors:**

See above.

**Limitations:**

yes

**Strengths And Weaknesses:**

Strengths:
- The paper provides theoretical guarantees that were lacking from Lu et al. (AAAI 2025). Bridging the gap between empirical performance and lack of theory.
- The stationary-point dichotomy is elegant: integral stationary points are local maximizers, while non-integral stationary points are strict saddles for λ > 1.
- They propose a saddle escaping Frank-Wolfe method, followed by empirical evaluation.

Weaknesses:
- Experiments are limited to validating convergence behavior and saddle escape, not end-to-end DkS solution quality against strong baselines.
- Perhaps I misunderstood, but wouldn't it make sense to compare your modified Frank-Wolfe algorithm to the original version? Preferably, on the entire DkS pipeline?

---

> ### Author Rebuttal · Authors · 2026-03-27
>
> Thank you for your review. Here are our response:
>
> >Experiments are limited to validating convergence behavior and saddle escape, not end-to-end DkS solution quality against strong baselines. Perhaps I misunderstood, but wouldn't it make sense to compare your modified Frank-Wolfe algorithm to the original version? Preferably, on the entire DkS pipeline?
>
> **Response:** We clarify that the primary goal of this paper is to provide a rigorous theoretical explanation for the empirical success of Frank–Wolfe in Lu et al. (2025). As discussed in Section 5.2, all non-integral stationary points are strict saddles with explicit positive curvature. It is our empirical observation from all real-world datasets used in Lu et al. (2025) that even the standard Frank–Wolfe with $\lambda = 1$ consistently escapes saddles and converges to an integral point in a finite number of steps. Therefore, the saddle-escaping mechanism is there for theoretical completeness and it is only triggered in "engineered" cases, such as the regular graph used in this paper. Comprehensive empirical comparisons against baselines on real-world datasets are already provided in Lu et al. (2025); repeating them here would not yield additional insight. The two experiments in this paper are specifically designed to verify the two key theoretical results: finite-step exact convergence and saddle escaping capability. We will include these clarifications in the final version.

---

> > ### Author Rebuttal · Reviewer_Y2Jg · 2026-04-01
> >
> > Thank you. My questions have been adequately answered.

---

### Official Review · Reviewer_X2mn · 2026-03-12

**Soundness:** 3
**Presentation:** 3
**Significance:** 3
**Originality:** 2
**Overall Recommendation:** 4
**Confidence:** 3

**Summary:**

This paper provides a theoretical analysis of a diagonal loading-based non-convex continuous relaxation for the Densest k-Subgraph problem. The authors resolve existing theoretical gaps by first proving that the relaxation is exactly tight for any arbitrary subgraph size k as long as the penalty parameter λ≥1. Furthermore, they fully characterize the underlying optimization landscape, establishing a strict dichotomy where all integral stationary points are local maximizers and all non-integral stationary points are strict saddle points with explicit positive curvature when λ>1 is non-integral. Leveraging this benign geometric property, the authors propose a deterministic Saddle-Escaping Frank-Wolfe (SE-FW) algorithm that efficiently escapes saddle points without relying on random perturbations, mathematically guaranteeing finite-step exact convergence to an integral local maximizer.

**Compliance With Llm Reviewing Policy:**

Affirmed.

**Final Justification:**

The author's response is reasonable, and I will maintain the weak accept rating.

**Key Questions For Authors:**

Please provide comments/responses on the weakness part.

**Strengths And Weaknesses:**

**Strengths**

S1. The paper closes a critical gap left by recent prior work. By rigorously proving that the continuous relaxation of the DkS problem is exactly tight for any arbitrary subgraph size k (provided λ ＞ 1) . This theoretical breakthrough significantly broadens the formulation's applicability to general subgraph mining scenarios.

S2. The authors proved the monotonicity of the local maximizers with respect to the penalty parameter λ brilliantly grounds the empirical observation that smaller penalty values yield more "friendly" optimization landscapes . The geometric analysis of the non-convex optimization landscape is original and insightful.

S3. The proposed SE-FW algorithm deterministically exploits explicit positive curvature to escape saddle points, avoiding the asymptotic limitations of standard random perturbations. Crucially, it achieves finite-step exact convergence to an integral local maximizer, completely eliminating the need for heuristic rounding. With an $O(m+n)$ per-iteration complexity, the method is highly scalable for real-world large graphs.

S4. This paper is logically clear, and the author explicitly lists the problems that have not yet been solved in existing literature and systematically answers them one by one, and the contribution of this paper can be clearly understood when reading.

**Weaknesses**

W1. The experimental section completely lacks performance comparisons against state-of-the-art DkS algorithms. The paper only compares the proposed SE-FW step-size rule against another Frank-Wolfe step-size rule . There is no evaluation against the exact prior penalty-based framework (Lu et al., 2025) or other methods. Cannot determine if SE-FW actually finds better results than existing methods.

W2. For a paper claiming that its O(m+n) iteration complexity makes it highly efficient for large-scale problems, the datasets used are far too small. The evaluation is restricted to a single SNAP Facebook graph with only 4,039 nodes and a synthetic 100-node regular graph.

W3. The paper notes that the penalty parameter λ shapes the optimization landscape , yet the experiments statically fix λ = 1.5 and use a single uniform initialization. The submission lacks crucial ablation studies analyzing how varying λ or using different random initializations impacts the convergence speed, the frequency of saddle escapes, and the final density of the extracted subgraphs.

W4. The paper focuses on the densest k-subgraph formulation. I am curious whether the proposed diagonal-loading formulation and relaxation could provide insights for related size-constrained dense subgraph variants, such as densest-at-least-k or densest-at-most-k. Prior work has studied these variants (e.g., [1] [2] [3]). A brief discussion on the applicability or limitations of the proposed approach in these settings would be helpful.

[1] R. Andersen and K. Chellapilla. Finding Dense Subgraphs with Size Bounds. WSDM, 2009.

[2] S. Khuller and B. Saha. On Finding Dense Subgraphs. ICALP, 2009.

[3] Y. Xu, et al. Efficient Algorithms for Densest-at-least-k Subgraph Discovery. SIGMOD, 2023.

---

> ### Author Rebuttal · Authors · 2026-03-27
>
> Thank you for your review. Here are our responses:
>
> **W1 & W2:** We clarify that the primary goal of this paper is to provide a rigorous theoretical explanation for the empirical success of Frank–Wolfe in Lu et al. (2025). As discussed in Section 5.2, all non-integral stationary points are strict saddles with explicit positive curvature. It is our empirical observation from all real-world datasets used in Lu et al. (2025) that even the standard Frank–Wolfe with $\lambda = 1$ consistently escapes saddles and converges to an integral point in a finite number of steps. Therefore, the saddle-escaping mechanism is there for theoretical completeness and it is only triggered in "engineered" cases, such as the regular graph used in this paper. Comprehensive empirical comparisons against baselines on real-world datasets are already provided in Lu et al. (2025); repeating them here would not yield additional insight. The two experiments in this paper are specifically designed to verify the two key theoretical results: finite-step exact convergence and saddle escaping capability. We will include these clarifications in the final version.
>
> **W3:** The sensitivity of solution quality to $\lambda$ has already been studied empirically in Lu et al. (2025), which provides an ablation across different values of $\lambda$ and demonstrates the overall trend of algorithm performance. Theorem 3.2 in this paper further provides a theoretical explanation for this empirical observation by proving that increasing $\lambda$ monotonically enlarges the set of local maximizers, making the optimization landscape progressively more challenging. Regarding initialization, we use uniform initialization because it injects no structural bias, which ensures that the algorithm's performance is driven entirely by the optimization landscape rather than a lucky initialization.
>
> **W4:** Both Dal$k$S and Dam$k$S involve fractional programming objectives, making direct extension of our framework non-trivial. A natural workaround is to enumerate $k$ and solve D$k$S for each $k$. Alternatively, another direction is to incorporate a subgraph size penalty term to cast the fractional programming objectives to quadratic programming objectives. Exploring the specific design of this penalty term requires careful consideration, and we leave this direction for future work.

---

> > ### Author Rebuttal · Reviewer_X2mn · 2026-04-01
> >
> > Thanks for the response. I'd like to maintain the weak accept rating.

---

### Official Review · Reviewer_7whr · 2026-03-13

**Soundness:** 3
**Presentation:** 3
**Significance:** 2
**Originality:** 2
**Overall Recommendation:** 4
**Confidence:** 4

**Summary:**

The paper studies a nonconvex, diagonally loaded continuous relaxation of the Densest k-Subgraph (DkS) problem and provides a largely theoretical contribution on (i) tightness of the relaxation for any k when λ ≥ 1; (ii) every integral stationary point is a local maximizer, and every non-integral stationary point is a strict saddle with explicit positive curvature (for λ > 1); and (iii) a saddle-escaping Frank–Wolfe (SE-FW) algorithm with a finite-step exact convergence guarantee to an integral local maximizer. The empirical section is limited and serves mainly to illustrate finite-step convergence and saddle escaping on small instances.

**Compliance With Llm Reviewing Policy:**

Affirmed.

**Final Justification:**

The rebuttal provided partial but convincing responses on W1, W4, and W5 as noted in my original review. The proofs are correct throughout, and the strict-saddle landscape characterization (Theorem 3.10) is a contribution that extends beyond D$k$S to other cardinality-constrained quadratic programs over polytopes.  The missing baselines (W3) and local-vs-global evidence (W2) remain real limitations.

**Key Questions For Authors:**

Q1. Please provide a precise statement of what is new in each of these results relative to these two prior works.
For example, Theorem 2.3 vs. Barman (SIAM J. Comput. 47(3), 2018, Lemma 5.1); Lemma 3.1 on integrality of local maximizers vs. the structural precedent in Hager and Krylyuk (SIAM J. Discrete Math. 12(4), 1999).

Q2. The convergence guarantee of SE-FW is to a local maximizer of an NP-hard problem. Can the authors characterize the quality of local maximizers relative to the global DkS optimum, even for restricted graph families such as the planted dense subgraph model or Erdős-Rényi graphs with a planted clique?

Q3. Can the authors state a single formal theorem synthesizing the total iteration count as a closed-form function of n, k, λ, and L? The components are proved separately in Theorems 4.1, 4.12 and Corollary 4.13;

Q4. Can the authors provide a direct empirical comparison of SE-FW solution quality and runtime against at minimum the Frank-Wolfe variant from the prior conference paper, which SE-FW directly extends? The current experiments establish that SE-FW converges to an integral solution, but not that it provides any practical improvement over the baseline algorithm it is designed to supplant.

Q5. The strict-saddle curvature bound and the tightness proof both depend on aᵢⱼ ∈ {0,1}. Is there a principled extension to weighted graphs, for instance with aᵢⱼ ∈ [0, W] for some W, that would require λ ≥ W rather than λ ≥ 1? If such an extension is non-trivial, please state this explicitly as a limitation.

**Limitations:**

The paper does not include a limitations section. The following should be addressed.

The convergence guarantee applies only to local maximizers. For an NP-hard problem this is a fundamental ceiling on what the theory provides, and the paper should explicitly acknowledge that SE-FW offers no approximation guarantee relative to the global optimum.

All proofs depend on aᵢⱼ ∈ {0,1}. The framework does not extend to weighted graphs without non-trivial modifications, and this should be stated rather than left implicit.

Theorem 3.7 requires λ to be non-integer. Whether the strict-saddle characterization holds for integer λ > 1, or whether degenerate stationary points that are neither local maximizers nor strict saddles can appear, is an open question that should be acknowledged.

**Strengths And Weaknesses:**

### Strengths

S1. The strict-saddle dichotomy (Theorems 3.7 and 3.10) is  original and a significant contribution. Proving that all non-integral stationary points are strict saddles with the explicit curvature bound 2(λ − aᵢⱼ) > 0 has no direct precedent in the DkS and conceptually transferable to other cardinality-constrained quadratic programs over polytopes.

S2. The finite-step exact convergence of SE-FW (Theorem 4.1, Corollary 4.13) closes a concrete open question left by the prior conference paper: the empirical observation that Frank-Wolfe always terminates at integral solutions. The deterministic escape mechanism, which exploits the explicit curvature bound rather than random perturbation  is a methodological contribution to the Frank-Wolfe literature on non-convex objectives (Lacoste-Julien, arXiv:1607.00345, 2016).


S3. The monotonicity result (Theorem 3.2) provides the first formal theoretical justification for the empirical observation in the prior conference paper that λ = 1 yields better solution quality than larger values. Establishing that increasing λ strictly enlarges the set of local maximizers is a clean and useful result for practitioners selecting this hyperparameter.


### Weaknesses

W1. The tightness result  is an incremental extension of existing work whose contribution may not meet the bar of a standalone theorem. The paper itself presents Barman (SIAM J. Comput. 47(3), 2018, Lemma 5.1) as Theorem 2.2 with λ = 2. Theorem 2.3 extends this to λ ≥ 1 via the single observation that aᵢⱼ ∈ {0,1} implies (λ − aᵢⱼ) ≥ 0 for any λ ≥ 1 in simple graphs, leaving the proof structure of Barman unchanged.

W2. The local-vs-global maximizer gap is entirely unaddressed and undermines the practical significance of the main algorithmic result. The paper guarantees convergence to a local maximizer of an NP-hard problem, yet provides no analysis of how close such solutions are to the global optimum and no empirical comparison of SE-FW solutions against best-known solutions on standard benchmark instances.

W3. The experimental section is insufficient for ICML. The two experiments (Facebook, n = 4,039; synthetic 10-regular, n = 100) verify the predicted theoretical phenomena but offer no comparison against any baseline, including the Frank-Wolfe variant from the prior conference paper that SE-FW directly extends, Liu et al. (arXiv:2511.11451, 2025), Konar and Sidiropoulos (WSDM 2021), or greedy peeling. There are no scalability experiments, no wall-clock times, no iteration counts, and no error bars. The penalty parameter λ is fixed at 1.5 throughout all experiments with no sensitivity analysis, despite Theorem 3.2 providing a theoretical basis for such a study.

W4. The paper does not provide its convergence results into a single formal total complexity theorem. The abstract positions finite-step convergence as a central claim

W5. The framework is restricted to simple unweighted graphs and this limitation is never acknowledged. The proofs throughout rely on aᵢⱼ ∈ {0,1} to ensure (λ − aᵢⱼ) ≥ 0 at λ ≥ 1. Whether the tightness guarantee, the strict-saddle characterization, or the curvature bound survives under edge weights is not discussed. Similarly, the non-integer requirement on λ in Theorem 3.7 is  never flagged as a structural assumption.

W6. Several presentation issues. The concepts and terms are used before they are defeined, e.g., the Frank-Wolfe gap G(x), the constraint set Cₖⁿ appears.

---

> ### Author Rebuttal · Authors · 2026-03-27
>
> Thank you for your review. Here are our responses:
>
> **W1 & Q1:** As acknowledged in our paper, Theorem 2.3 is indeed an extension of Barman's work. However, lowering the requirement from $\lambda=2$ to $\lambda\ge1$ represents a significant theoretical milestone. Specifically, Lu et al. (2025) prove that $\lambda\ge1$ is a necessary condition for tightness. Theorem 2.3 establishes that it is also sufficient, closing the theoretical gap. Furthermore, finding this exact tight bound is crucial for the optimization landscape and algorithmic design, as Theorem 3.2 proves that over-penalizing monotonically enlarges the set of local maximizers.
>
> As explicitly cited in our paper, Hager & Krylyuk (1999) inspired our Theorem 3.6 (not Lemma 3.1). Although the condition in our Theorem 3.6 is mathematically equivalent to that in their Corollary 3.2, our condition is much simpler, as it directly applies to elements of the gradient and this is conducive and revealing for the subsequent analysis.
>
> **W2 & Q2 & L1:** As shown in Jones et al. (2023), it is conjectured impossible for an efficient algorithm to achieve an $O(n^{1/4-\epsilon})$ approximation. Such approximation ratios are too pessimistic to explain the empirical success in Lu et al. (2025). Hence, we aim to rigorously explain the empirical success of the continuous non-convex relaxation via landscape and convergence analysis for general graphs. Moreover, finite-step exact convergence to an integral local maximizer is a much stronger guarantee than the asymptotic convergence to a stationary point provided by standard first-order methods for non-convex optimization.  For restricted models like the planted clique, Theorem 3.6 can be used to characterize spurious local maximizers. We leave this direction as future work.
>
> **W3 & Q4:** We clarify that the primary goal of this paper is to provide a rigorous theoretical explanation for the empirical success of Frank–Wolfe in Lu et al. (2025). As discussed in Section 5.2, all non-integral stationary points are strict saddles with explicit positive curvature. It is our empirical observation from all real-world datasets used in Lu et al. (2025) that even the standard Frank–Wolfe with $\lambda=1$ consistently escapes saddles and converges to an integral point in a finite number of steps. Therefore, the saddle-escaping mechanism is there for theoretical completeness and it is only triggered in "engineered" cases, such as the regular graph used in this paper. Comprehensive empirical comparisons against baselines on real-world datasets are already provided in Lu et al. (2025); repeating them here would not yield additional insight. The two experiments in this paper are specifically designed to verify the two key theoretical results: finite-step exact convergence and saddle escaping capability. We will include these clarifications in the final version.
>
> **W4 & Q3:** To trigger a saddle escape, the standard Frank–Wolfe steps must reduce the Frank–Wolfe gap to $\epsilon_1=\min\\{\frac{\xi_y^2}{16(\lambda+1)^2\tau^2n^2},\theta\\}$. By Equations (37) and (38), each iteration guarantees an ascent of at least $\min\\{\frac{\epsilon_1^2}{4kL},\frac{\epsilon_1}{2}\\}$, and the maximum objective capacity is $O(k^2)$, the total number of standard Frank–Wolfe steps before the last saddle escape is at most $O(\max\\{\frac{k^3L}{\epsilon_1^2},\frac{k^2}{\epsilon_1}\\})$. By Corollary 4.13, the number of saddle escapes is at most $O(k^2n^2)$. Afterward, final standard Frank–Wolfe steps to enter a local maximizer's basin require reducing the gap to $\epsilon_2=\min\\{\frac{\sigma_x^2}{16\tau^2L^2},\theta\\}$. The number of iterations for this is at most $O(\max\\{ \frac{k^3L}{\epsilon_2^2},\frac{k^2}{\epsilon_2}\\})$. Therefore, the total number of iterations is at most $O(\max\\{\frac{k^3L}{\epsilon_1^2},\frac{k^2}{\epsilon_1}\\}+k^2n^2+\max\\{\frac{k^3L}{\epsilon_2^2},\frac{k^2}{\epsilon_2}\\})$. We will consolidate these together in the final version.
>
> **W5 & Q5 & L2 & L3:** Our  framework naturally extends to weighted graphs by setting $\lambda\ge w_\max$, where $w_\max$ denotes the maximum edge weight. The algorithm and most of the theoretical analysis can be extended to the weighted case with moderate modifications. We chose to focus on unweighted graphs because that is the standard form of D$k$S (Feige et al., 2001; Bhaskara et al., 2010).
>
> $\lambda$ is a user-defined hyperparameter rather than a structural assumption. Our landscape analysis implies that $\lambda$ slightly above 1 is the best choice. By Theorem 3.2, increasing $\lambda$ monotonically enlarges the set of local maximizers, so there is no reason to choose an integral value of $\lambda>1$. Furthermore, the strict saddle characterization established in Theorem 3.10 only requires $\lambda>1$ and does not impose any non-integral condition on $\lambda$.
>
> **W6:** Thank you for pointing them out. We will fix all presentation issues in the final version.

---

> > ### Author Rebuttal · Reviewer_7whr · 2026-04-03
> >
> > Thank you for the detailed rebuttal. The response helped clarify several points, and in particular the iteration bound composition (W4) and the non-integer $\\lambda$ clarification (W5) were informative. I am raising my score from 3 to 4 because the theoretical contributions are stronger than I initially credited. I select (c) because the two remaining concerns, missing baselines (W3) and the local-vs-global gap (W2), are about empirical evaluation and equire new experiments or analysis in a revised paper.
> >
> > ### Fully resolved
> >
> > **W1/Q1 (Tightness).**
> >
> > **W4/Q3 (Total iteration complexity).**
> >
> > **W5/Q5 (Non-integer $\\lambda$, weighted graphs).**
> >
> > **W6 (Presentation).** Acknowledged.
> >
> > ### Remaining Issues
> >
> > **W3/Q4 (No baselines).** This remains a significant open issue. SE-FW is a new algorithm that differs from the standard FW in Lu et al. (2025) in its step-size rule, penalty parameter ($\\lambda = 1.5$ versus 1), and escape mechanism. Pointing to Lu et al.'s experiments with a different algorithm at a different parameter does not establish how SE-FW performs. The rebuttal's observation that standard FW with $\\lambda = 1$ already escapes saddles on all real-world datasets is interesting, but it raises a question: if the escape mechanism is never triggered in practice, SE-FW reduces to standard FW on real graphs, and the algorithmic contribution is a theoretical guarantee rather than a practical improvement. The paper's framing is already largely theoretical, and I do not view this as disqualifying, but a minimal comparison confirming that SE-FW matches standard FW in solution quality, with iteration counts and runtimes, would strengthen rather than weaken the theoretical narrative. I note that this concern was raised independently by multiple reviewers in the initial round, so it is not an isolated observation.
> >
> > Relatedly, a scope gap sharpens the concern I raised in W3: the paper claims to explain the empirical success of Frank–Wolfe in Lu et al. (2025), but Lu et al. used $\\lambda = 1$, where the landscape theory (Theorems 3.7 and 3.10, which require $\\lambda > 1$) provides no guarantees.
> >
> > **W2/Q2 (Local-vs-global gap).** The Jones et al. (2023) hardness result is useful context for why worst-case approximation guarantees are out of reach. However, worst-case hardness does not preclude instance-specific analysis. My Q2 asked about restricted graph families, planted dense subgraph models, Erdős–Rényi with planted cliques, and the rebuttal acknowledges Theorem 3.6 could be applied there but defers entirely to future work. Even a small empirical measurement of solution quality versus known optima on small planted instances would go a long way toward demonstrating that the local maximizers found are meaningful.

---

> > > ### Author Response · Authors · 2026-04-07
> > >
> > > Thank you for your acknowledgment and follow-up questions. Here are our responses:
> > >
> > > **W3/Q4 (No baselines):** Thank you for the suggestion. We evaluated the performance of SE-FW with $\lambda=1.5$ and standard FW with $\lambda=1$ using uniform initialization on the Facebook and web-Stanford datasets. The performance comparison is summarized in the following table. It is evident that both algorithms exhibit near identical empirical performance. We will include more comparisons between SE-FW and standard FW in the final version.
> > >
> > > | Dataset | $k$ | Algorithm | Iterations | Density |
> > > | :--- | :---: | :--- | :---: | :---: |
> > > | Facebook | 20 | Standard FW | 35 | 1.0 |
> > > | Facebook | 20 | SE-FW | 36 | 1.0 |
> > > | Facebook | 50 | Standard FW | 15 | 1.0 |
> > > | Facebook | 50 | SE-FW | 15 | 1.0 |
> > > | web-Stanford | 20 | Standard FW | 110 | 0.9895 |
> > > | web-Stanford | 20 | SE-FW | 110 | 0.9895 |
> > > | web-Stanford | 50 | Standard FW | 67 | 0.7298 |
> > > | web-Stanford | 50 | SE-FW | 66 | 0.7298 |
> > >
> > > Regarding the gap between our theoretical results and empirical results in Lu et al. (2025), Theorems 3.7 and 3.10 indeed require $\lambda>1$. However, our landscape analysis still provides some insight into the empirical success observed at $\lambda=1$. In particular, Theorem 3.4 establishes necessary and sufficient conditions for stationary points, which holds for $\lambda=1$ as well. Consider a non-integral stationary point under $\lambda=1$. For such a point to have no strict ascent direction, we need $a_{ij}=1$ for every pair of distinct indices $i,j\in\mathcal{S}_{f}$, that is, every pair of distinct indices in the non-integer index set must be connected by an edge. Otherwise, the quadratic term remains positive, implying the existence of a strict ascent direction. Given that real-world graphs are typically sparse, such stationary points are typically strict saddle points even when $\lambda=1$, making it unlikely for the algorithm to be trapped in their vicinity. We will include this discussion in the final version.
> > >
> > > **W2/Q2 (Local-vs-global gap):** Thank you for your suggestion. In order to answer your question, we generated 20 Erdős–Rényi random graphs with a planted clique (20 consecutive random seeds from 0 to 19) with $n=10,000$, $p=0.05$, and $k=30$. We ran greedy peeling and SE-FW with $\lambda=1.5$ and uniform initialization for each of these 20 problem instances. The results were as follows. Greedy peeling recovered the planted clique in 2/20 instances, while SE-FW succeeded in 9/20 instances (succeeded on both instances where greedy peeling succeeded and the saddle escaping mechanism never occurred). These results further validate the effectiveness of our framework. We will add this experiment in the final version.

---

### Decision · Program_Chairs · 2026-04-30

**Decision:**

Accept (regular)

**Comment:**

This paper provides a rigorous theoretical analysis of a recent diagonal-loading non-convex relaxation for the Densest $k$-Subgraph problem, successfully characterizing its stationary landscape and proposing a novel saddle-escaping Frank-Wolfe algorithm.

Strengths
- Resolves an open problem introduced in recent prior work (Lu et al., AAAI 2025).
- Introduces a novel, theoretically grounded algorithmic variant based on the Frank-Wolfe method.

Weaknesses
- Experimental validation is limited, though acceptable given theoretical focus

Decision
Accept (Poster)